# BENCHMARKING UNLEARNING FOR VISION TRANS-FORMERS

## ABSTRACT

Research in machine unlearning (MU) has gained strong momentum: MU is now widely regarded as a critical capability for building safe and fair AI. In parallel, research into transformer architectures for computer vision tasks has been highly successful: Increasingly, Vision Transformers (VTs) emerge as strong alternatives to CNNs. Yet, MU research for vision tasks has largely centered on CNNs, not VTs. While benchmarking MU efforts have addressed LLMs, diffusion models, and CNNs, none exist for VTs. *This work is the first to attempt this, benchmarking MU algorithm performance in different VT families (ViT and Swin-T) and at different capacities.* The work employs (i) different datasets, selected to assess the impacts of dataset scale and complexity; (ii) different MU algorithms, selected to represent fundamentally different approaches for MU; and (iii) both single-shot and continual unlearning protocols. Additionally, it focuses on benchmarking MU algorithms that leverage training data memorization, since leveraging memorization has been recently discovered to significantly improve the performance of previously SOTA algorithms. En route, the work characterizes how VTs memorize training data relative to CNNs, and assesses the impact of different memorization proxies on performance. The benchmark uses unified evaluation metrics that capture two complementary notions of forget quality along with accuracy on unseen (test) data and on retained data. Overall, this work offers a benchmarking basis, enabling reproducible, fair, and comprehensive comparisons of existing (and future) MU algorithms on VTs. And, for the first time, it sheds light on how well existing algorithms work in VT settings, establishing the current state of affairs.

## 1 INTRODUCTION

The high success of Vision Transformers (VTs) brings with it new responsibilities regarding responsible/fair/safe AI. Among these, the ability to remove (the influence of) specific "problematic" data from trained models (a.k.a. machine unlearning) (MU) is critical. Said problematic data may include biased, erroneous, poisoned, obsolete, or privacy-sensitive data. Popular VT architectures represent thus an important frontier in this challenge. On the other hand, recently memorization has been identified as playing a fundamental role for MU (and this holds across modalities). For instance, MU research in LLMs (Barbulescu and Triantafillou, 2024; Jang et al., 2022) and in diffusion models (Ren et al., 2024; Wen et al., 2024) explicitly detects and mitigates memorization. In computer-vision tasks, memorization has been shown to be a key factor affecting unlearning performance (Zhao et al., 2024; Torkzadehmahani et al., 2024).

Despite the fact that memorization and MU have been extensively studied in LLMs, text-to-image Diffusion Models, and Convolutional Networks (CNNs), it is an open question whether findings transfer to VTs. This uncertainty stems from key differences in architecture, training regimes, and inductive biases. Compared to LLMs (with which they share a transformer backbone), VTs operate on image patches using global self-attention (unlike LLMs which rely on causal masking or token-order constraints and process semantically meaningful tokens) and VTs are typically (pre)trained using supervised classification objectives (e.g., on ImageNet), while LLMs are trained on large corpora using self-supervised objectives, yielding different types of data exposure and memorization. Compared to CNNs, VTs lack strong spatial inductive biases such as locality (semantically related neighbouring pixels), translation equivariance (for positional reasoning). These biases help CNNs localize and isolate memorization effects, whereas VTs must learn such structure from data alone,

making VTs more data-hungry, often necessitating a pretrain–then–finetune training regime. Moreover, unlike LLMs, VTs lack the benefit of language's syntactic and semantic structure. Hence, learned representations are more entangled and spread across layers and attention heads.

**The Gap.** One can thus reasonably expect the above differences to collectively introduce unique challenges for unlearning in VTs. Said potential challenges remain largely unexplored, despite a few related research works on both the algorithmic side (e.g., Cadet et al. (2024); Cho et al. (2024) whose evaluations of CNN-derived unlearning also include some VT, typically, ViT-Tiny) and on the benchmarking side, with recent efforts systematically evaluating MU across tasks and modalities (such as Maini et al. (2024); Li et al. (2024) for LLMs, Ma et al. (2024); Zhang et al. (2024), and Cheng and Amiri (2024) for text-to-image diffusion models).

Focusing on vision, image classification tasks, the first comprehensive attempt in this domain, by Triantafillou et al. (2024), was based on the NeurIPS 2023 MU competition, evaluating and ranking the top algorithms on two datasets from the competition on a CNN (ResNet-18) architecture. Interestingly, they report that MU methods can be brittle across architectures/datasets. Grimes et al. (2024) benchmarked MU on the same CNN environment but focused on evaluating more demanding privacy threats and paid attention to continual MU. Cadet et al. (2024) extended the benchmark in Triantafillou et al. (2024), evaluating 18 MU algorithms on four datasets on a ResNet (as well as on ViT-Tiny). Each of the above benchmarks has a different emphasis: Each is typically designed for a specific architecture-modality-task pairing. Some stress new datasets, while others stress comprehensive/exhaustive examinations/rankings of a large set of MU algorithms.

**This work.** Overall, benchmarking for MU has been adressed for LLMs, diffusion models, and CNNs, but not for VTs. We fill this gap by benchmarking MU on VTs, contributing an evaluation basis along the following key axes:

1. **Memorization:** Do VTs memorize differently to CNNs? How does this affect unlearning?
2. **Proxies:** Are CNN-based memorization proxies effective in VTs? Can they improve unlearning?
3. **Algorithms:** How do fundamentally different approaches for CNN-based MU perform on VTs?
4. **VT architectures:** Do VTs design choices (ViT vs. Swin-T) and *capacity* impact unlearning?
5. **Pretrain–Finetune:** How does the VTs' pretrain–finetune paradigm influence MU?
6. **Algorithm-Architecture Pairings:** Are certain pairings especially compatible?
7. **Continual MU on VTs:** How stable is performance under continual unlearning?

In contrast to prior "standard" benchmarking research which are leaderboard-style (benchmarking/ranking a large set of algorithms) or proposing new datasets, our work follows a different path: It is centered on VTs, systematically testing *representative, memorization-leveraging* MU approaches across *two* VT families (ViT, Swin-T) at *two* capacities each, over *four* datasets of varying size/complexity, and under *both* single-shot and continual unlearning scenarios. We employ unified metrics (ToW, ToW-MIA) that jointly account for retain/test accuracy, forget accuracy, and MIA vulnerability. We contrast results against CNN-derived MU counterpart algorithms. We aim to isolate architecture, capacity, memorization (and proxy) effects to assess how well MU methods perform in VTs and establish the current state of affairs regarding MU performance in VTs. The code for reproducing the results is available at: `https://anonymous.4open.science/r/unlearning_VTs-31E1`.

## 2 MODEL ARCHITECTURES AND ALGORITHMS

### 2.1 VISION TRANSFORMERS

Transformer architectures (Vaswani et al., 2017) leverage self-attention mechanisms instead of recurrence. We focus on, arguably, the two most popular VTs: ViT and Swin-T. ViT (Dosovitskiy et al., 2021) processes images as sequences of flattened, embedded patches, which are (i) projected into an embedding space, (ii) combined with positional encodings, and (iii) passed through transformer encoder layers comprising multi-headed self-attention (MSA) and multi-layer perceptrons (MLP). Swin-T (Liu et al., 2021) adds hierarchical representations (a la CNNs) by progressively merging patches and reducing spatial resolution at deeper layers, forming multi-scale representations. Additionally, a shifted-window self-attention mechanism is introduced to model both local and global contexts. These allow us to study MU performance on architectures which, due to having/lacking hierarchical representations, are more/less similar to CNNs (Swin-T/ViT).

## 2.2 MACHINE UNLEARNING (MU)

MU (Cao and Yang, 2015) aims to remove the influence of specific ("problematic") training examples from pretrained models. A large body of work has tackled MU, spanning formal definitions and guarantees (Ginart et al., 2019; Sekhari et al., 2021; Neel et al., 2020), empirically effective methods (Goel et al., 2022; Golatkar et al., 2020; Thudi et al., 2022), and other earlier efforts (Xu et al., 2023). Over time, several key baselines have emerged, capturing fundamental components for successful MU. These range from simpler baselines like Fine-tune (FT) (Warnecke et al., 2023; Golatkar et al., 2020), to more sophisticated baselines like NegGrad+ (Kurmanji et al., 2023). Most recent SOTA algorithms include SCRUB (Kurmanji et al., 2023), L1-sparse (Jia et al., 2024), Salun (Fan et al., 2024) and the meta-algorithmic framework RUM (Zhao et al., 2024).

Evidently, all of the above methods have focused on CNNs. It is noted that, contemporaneously, LetheViT (Tong et al., 2025) and NOVO (Roy et al., 2025) studied unlearning specifically for VTs. Our contribution is orthogonal: we provide an architecture- and capacity-aware benchmark for CNN-derived, memorization-based MU algorithms on VTs across families, sizes, datasets, and protocols. As these works are **contemporaneous** (and not peer-reviewed yet), we do not directly compare against their results. Nonetheless, our benchmark offers a substrate where such new methods and architectures can be integrated and systematically evaluated.

Following guidance from Zhao et al. (2024), we focus on three representative methods: (i) FT as a standard baseline, effective for low-memorization examples; (ii) NegGrad+ as a high-performing advanced baseline, and excels when leveraging memorization; and (iii) SalUn as a recent SOTA. Each is instantiated within RUM, which has been shown to strengthen the above (and other SOTA) algorithms.

MU algorithms operate assuming a training data set $D$ which is split into a forget set $D_f$ (comprising the examples to be forgotten) and a retain set $D_r$, with $D_r = D \setminus D_f$. FT continues training exclusively on $D_r$. NegGrad+ promotes forgetting through gradient ascent (GA) on $D_f$, while simultaneously performing gradient descent on $D_r$ optimizing:

$$L(\theta) = \frac{\beta}{|D_r|} \sum_{(x_i, y_i) \in D_r} l(f(x_i; \theta), y_i) - \frac{(1-\beta)}{|D_f|} \sum_{(x_j, y_j) \in D_f} l(f(x_j; \theta), y_j), \quad (1)$$

where $\beta \in [0, 1]$ balances retention and forgetting. Thus, NegGrad+ integrates FT with GA. (NB: GA and variants (Jang et al., 2022; Barbulescu and Triantafillou, 2024) have been shown to work surprisingly well for LLMs, but not for CNNs (Kurmanji et al., 2023)). Finally, SalUn uses a saliency score to identify the key parameters influencing examples in $D_f$. The loss function $L(\theta)$ replaces forget example labels $y_j$ with random ones $y_j'$ and incorporates a regularization parameter $\alpha$. During optimization of $L(\theta)$, only the salient weights are updated.

$$L(\theta) = \frac{\alpha}{|D_r|} \sum_{(x_i, y_i) \in D_r} l(f(x_i; \theta), y_i) + \frac{1}{|D_f|} \sum_{(x_j, y_j) \in D_f} l(f(x_j; \theta), y_j') \quad (2)$$

A key consideration for our selections is the fact that they represent different unlearning paradigms. NegGrad+ updates gradients globally (all model weights), whereas SalUn first computes the salient weights for $D_f$ examples and then updates only them by adding "noise".

## 2.3 LEVERAGING MEMORIZATION FOR UNLEARNING

Memorization quantifies model dependency on specific examples (Feldman, 2019; Feldman and Zhang, 2020). The measure, referred to as *memorization score*, is defined as:

$$\text{mem}(A, D, i) := \Pr_{f \sim A(D)}[f(x_i) = y_i] - \Pr_{f \sim A(D \setminus \{(x_i, y_i)\})}[f(x_i) = y_i], \quad (3)$$

capturing how predictions change when removing an example from training. Robust computation of this metric necessitates training large number of models (each retrained on subsets excluding specific examples) to achieve stable estimates. This is especially expensive, particularly for VTs. Proxies like the four *Learning Events Proxies* (Confidence (C), Max Confidence (MaxC), Entropy (E), and

Binary Accuracy (BA)) (Jiang et al., 2021) and *Holdout Retraining (HR)* (Carlini et al., 2019) come to the rescue, efficiently estimating memorization scores. The formulas for each of these, for space reasons, can be found in the Appendix A.2.

RUM leverages memorization to improve unlearning performance. It is structured in three stages:

1. Refinement: Partitions the forget set into homogeneous subsets based on memorization scores of examples in the forget set. Specifically, it creates three partitions consisting of low-, medium- and highly-memorized forget examples.
2. Matching: Selects suitable unlearning methods tailored to each partition.
3. Unlearning: Applies an unlearning algorithm sequentially on the three partitions.

We employ RUM for each of FT, NegGrad+, and SalUn. RUM(SalUn) applies the SalUn algorithm sequentially to the three partitions. Likewise for RUM(NegGrad+), and RUM(FT). We refer to NegGrad+, SalUn, or FT, for simplicity, instead of RUM(NegGrad+), RUM(SalUn), and RUM(FT).

## 3 EXPERIMENTAL SETUP

### 3.1 DATA SETS

We use four different datasets, varying in terms of semantic complexity, size, and number of classes. We employ CIFAR-10 (medium complexity) and CIFAR-100 (high complexity) (Krizhevsky, 2009) as well as SVHN (Netzer et al., 2011) (larger with lower complexity, as objects are digits). These are standard datasets against which MU algorithms have been evaluated. CIFAR-10 consists of 60K RGB color images, each of 32×32 pixels, with 10 classes. CIFAR-100 shares the same format but has 100 classes (each with 600 images). SVHN consists of 600K+ images of digits, each of 32×32 pixels, with 10 classes. ImageNet-1K (Deng et al., 2009) serves as the foundation for pretraining VTs. It contains 1.3M images across 1000 classes. In addition, we also used the ImageNet-1K validation set (50K images) to benchmark against larger, more complex datasets.

### 3.2 TRANSFORMER ARCHITECTURES AND IMPLEMENTATIONS

We selected ViT and Swin-T variants similarly sized to ResNet-50 (ca. 25.6 million parameters) for fair comparisons against CNNs. The main ViT-Small variant contains ca. 21.6 million parameters. It uses $8 \times 8$ pixel patch sizes. The main Swin-Tiny variant has ca. 27.5 million parameters and uses a patch size of $4 \times 4$ pixels and a window size of $7 \times 7$. These VTs serve as our main benchmark architectures because: (i) they are similar-sized models in their respective families meant for comparable applications; (ii) their inductive biases place them closer (Swin-T) or further (ViT) to/from CNNs. By being similarly-sized, our results reveal whether the hierarchical or locality biases of Swin-T play a role in unlearning. We also employ ViT-Tiny (ca. 5.5 million parameters) and Swin-Small (ca. 48.8 million parameters) to reveal model capacity effects.

### 3.3 EVALUATION METRICS

UNLEARNING METRICS

The unlearned model $\theta_u = U(\theta_o, D_f, D_r)$ must balance forgetting quality (on $D_f$) while preserving model performance (on $D_r$) and generalizing well to unseen test data $D_{test}$. We adopt two primary metrics from Zhao et al. (2024) to assess this balance. ToW$(\theta_u, \theta_r, D_f, D_r, D_{test})$ (ToW for short):

$$\text{ToW} = (1 - \Delta a(\theta_u, \theta_r, D_f)) \cdot (1 - \Delta a(\theta_u, \theta_r, D_r)) \cdot (1 - \Delta a(\theta_u, \theta_r, D_{test})) \quad (4)$$

where $a(\theta, D) = \frac{1}{|D|} \sum_{(x,y) \in D} \mathbb{1}[f(x; \theta) = y]$ is the accuracy of a model $f$ parametrized by $\theta$ on $D$, and $\Delta a(\theta_u, \theta_r, D) = |a(\theta_u, D) - a(\theta_r, D)|$ is the absolute difference in accuracy between $\theta_u$ and the retrained-from-scratch model $\theta_r$ on $D$. And, ToW-MIA$(\theta_u, \theta_r, D_f, D_r, D_{test})$ (ToW-MIA):

$$\text{ToW-MIA} = (1 - \Delta m(\theta_u, \theta_r, D_f)) \cdot (1 - \Delta a(\theta_u, \theta_r, D_r)) \cdot (1 - \Delta a(\theta_u, \theta_r, D_{test})) \quad (5)$$

where $m(\theta, D) = \frac{TN_D}{|D|}$ and $\Delta m(\theta_u, \theta_r, D) = |m(\theta_u, D) - m(\theta_r, D)|$. In ToW-MIA, $m(\theta, D)$ accounts for "forget quality" using Membership Inference Attack (MIA) performance. To calculate this, as in Fan et al. (2024); Zhao et al. (2024); Jia et al. (2023), we train a binary classifier $C$ that

classifies examples as "in-training" or "out-of-training". We simplify our notation $l(f(x;\theta),y)$ (the cross-entropy loss of a model with weights $\theta$ on example $x$ with label $y$) to $l(x,y)$. $C$ is trained on loss values from a balanced dataset $D_{\mathsf{t}}^b = \{(l(x_i,y_i),y_i^b)\}$, where examples $x_i$ are drawn equally from the retain set $D_r$ (labelled as $y_i^b = 1$ for "training") and the test set $D_{\text{test}}$ (labelled as $y_i^b = 0$ for "non-training"). Once trained, $C$ evaluates loss values for examples in $D_f$. $m(\theta, D)$ measures the proportion of $D_f$ examples that $C$ classified as "non-training", i.e. the true negatives $TN_D$. The hope is for the unlearning method to cause the model to treat forget set examples as if they were never seen during training, resulting in the classifier categorising them as "non-training." So similarly to $\Delta a$, $\Delta m(\theta_u, \theta_r, D)$ quantifies how closely the unlearned model $\theta_u$ resembles the MIA performance of the retrain-from-scratch model $\theta_r$.

ToW and ToW-MIA measure "forget quality" differently: ToW uses accuracy differences on the forget set, while ToW-MIA uses differences in MIA vulnerability. Both range from 0 to 1, with higher values indicating better unlearning performance (closer-matching retraining from scratch). Together, ToW and ToW-MIA provide a comprehensive view of unlearning performance.

### MEMORIZATION PROXY METRIC

We compute Spearman's rank correlation coefficient ($\rho$) between the memorization score and the proxy values. For a training example $(x_i, y_i) \in D$, we define: $m_i = \text{mem}_m(A, D, i)$ as the "Feldman" memorization score and $p_i = \text{proxy}(x_i, y_i)$ as the value from one of our proxy metrics. We rank these values in ascending order to obtain the corresponding ranks $\text{R}(m_i)$ and $\text{R}(p_i)$. Then we compute $\rho$:

$$\rho = \frac{\sum_{i=1}^{|D|}(\text{R}(m_i) - \overline{\text{R}(m)}) \cdot (\text{R}(p_i) - \overline{\text{R}(p)})}{\sqrt{\sum_{i=1}^{|D|}(\text{R}(m_i) - \overline{\text{R}(m)})^2 \cdot \sum_{i=1}^{|D|}(\text{R}(p_i) - \overline{\text{R}(p)})^2}} \tag{6}$$

Here, $\overline{\text{R}(m)}$ and $\overline{\text{R}(p)}$ denote the mean ranks of the memorization scores and proxy values respectively and $|D|$ is the dataset size. The resulting coefficient $\rho \in [-1, 1]$ indicates the strength and direction of the monotonic relationship between the memorization score and the proxy.

## 4 RESULTS AND ANALYSES

In this section we present the main results, focusing on CIFAR-100 as a representative dataset. The Appendix contains results for the other datasets. Also, as we focus on memorization-based MU algorithms, we first established that (i) the memorization patterns in VTs mimic those in CNNs and that (ii) memorization proxies can indeed be trustworthy for being leveraged by MU algorithms in VTs. The detailed results can be found in Appendix A.2.

### 4.1 MEMORIZATION AND PROXIES IN VTS

Memorization plays a central role in unlearning, so we first ask whether VTs memorize similarly to CNNs, and whether CNN-derived proxies remain valid predictors of memorization in VTs.

**Key Takeaways–Memorization patterns.** We observe (see Figure 3, 4 in Appendix A.2.1) the same long-tailed distributions previously reported for CNNs (Feldman and Zhang, 2020). Thus, VTs and CNNs display fundamentally similar memorization behavior, despite architectural differences. Also, on CIFAR-10, VTs exhibit slightly lower memorization than ResNet-18, reflecting their ability (via pretraining and global attention) to rely less on memorizing individual examples for simpler tasks.

**Key Takeaways–Proxy validity.** Whether CNN-derived proxies remain valid for VTs is unclear due to differences in their inductive biases and training regimes (finetune-then-retrain). We evaluated five memorization proxies: Confidence, Max Confidence, Entropy, Binary Accuracy (Jiang et al., 2021), and Holdout Retraining (Carlini et al., 2019). The results are in Table 1.

The Confidence proxy consistently achieves the strongest correlations across all models and datasets, with magnitudes (-0.79 to -0.91) closely resembling those in CNNs. Swin-Tiny shows slightly stronger correlations than ViT-Small, likely due to its hierarchical structure that more closely resembles traditional CNNs. Holdout Retraining shows moderate but significant positive correlations, and is attractive in practice given its large computational advantages vis-a-vis the other proxies. Over-

all, simple proxies such as Confidence and Holdout Retraining remain predictive for VTs, enabling scalable memorization-based unlearning without expensive Feldman-score compuations.

Table 1: Spearman Correlation Coefficients Between Memorization and Proxies

| | CIFAR-10 | | | CIFAR-100 | | |
|---|---|---|---|---|---|---|
| **Proxy** | **ResNet-18** | **ViT-Small** | **Swin-Tiny** | **ResNet-50** | **ViT-Small** | **Swin-Tiny** |
| **Confidence** | -0.80 | -0.79 | -0.88 | -0.91 | -0.85 | -0.90 |
| Max Confidence | -0.76 | -0.77 | -0.85 | -0.87 | -0.80 | -0.86 |
| Entropy | -0.75 | -0.78 | -0.85 | -0.80 | -0.77 | -0.82 |
| Binary Accuracy | -0.71 | -0.63 | -0.79 | -0.89 | -0.69 | -0.78 |
| **Holdout Retraining** | +0.67 | +0.45 | +0.64 | +0.62 | +0.50 | +0.52 |

## 4.2 Unlearning Algorithms Performance on Vision Transformers

For these experiments, $D_f$ comprises 3,000 examples, divided into $M = 3$ partitions of $N = 1,000$ examples each, representing the lowest, medium and highest proxy values. We apply each algorithm unlearning in the order of low $\rightarrow$ medium $\rightarrow$ high memorization, using memorization-proxy values. All results are averaged over three runs and we also present 95% confidence intervals.

Given that pre-trained transformer models have lower memorization, we anticipated that $\theta_r$ might already perform well on $D_f$. To establish a baseline for comparison, we calculated ToW and ToW-MIA parametrized by $(\theta_o, \theta_r, D_f, D_r, D_{test})$ denoted as "Original". This baseline indicates the performance we would achieve without applying any unlearning algorithm. Hyper-parameters for all algorithms are detailed in Appendix A.1.2.

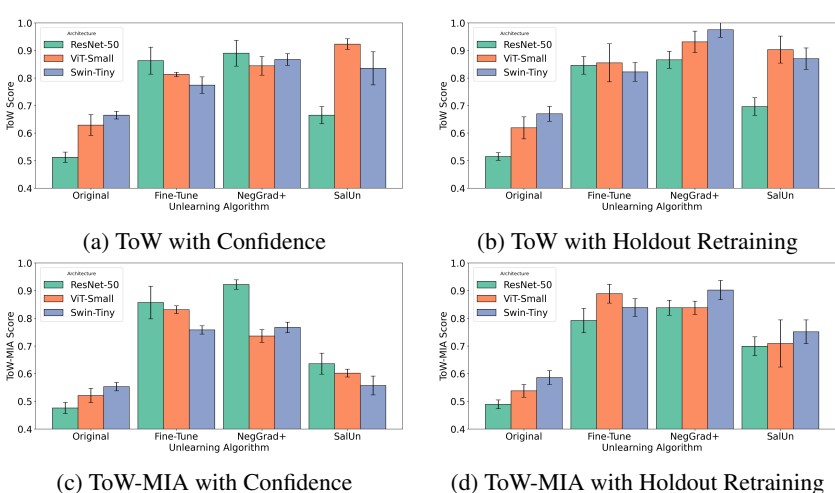

(a) ToW with Confidence

(b) ToW with Holdout Retraining

(c) ToW-MIA with Confidence

(d) ToW-MIA with Holdout Retraining

Figure 1: MU performance comparison on CIFAR-100

**How do different MU approaches perform on VTs?** Figures 1 present unlearning results across both VTs and CIFAR-100 (see Appendix Figure 5 and 7 for results on CIFAR-10 and SVHN datasets, and Table 11 for detailed data). Our results show that simple approaches like Fine-tune perform surprisingly well, especially on the simpler SVHN dataset. NegGrad+ is consistently strong in all cases, and especially for more complex datasets (CIFAR-100) and outperforms all when paired with Holdout Retraining. Notably, NegGrad+ with Holdout Retraining for Swin-T shows even better performance than in ResNets. SalUn achieves good ToW scores but struggles with ToW-MIA on more complex datasets (CIFAR-100), though it performs competitively on SVHN. From the proxy viewpoint, Holdout Retraining performs excellently compared to Confidence for CIFAR-10/CIFAR-100, while Confidence can match it for the simpler SVHN.

**Key Takeaways.** SOTA MU algorithms from CNNs can be equally (if not more) effective for VTs. NegGrad+ is the most robust MU method for VTs, while SalUn is vulnerable in ToW-MIA on harder datasets. Proxy choice (Holdout Retraining vs. Confidence) further shapes outcomes.

**How does unlearning performance in VTs compare to CNNs?** We first compare the accuracy of the retrained model $\theta_r$ on $D_f$ in ResNets versus VTs, since this directly influences ToW and ToW-

MIA evaluation for the benchmark model $\theta_r$. Table 2 shows that on CIFAR-10, $\theta_r$ achieves much higher $D_f$ accuracy in VTs than CNNs (e.g., $> 90\%$ for ViT/Swin-T vs. $\sim 50\%$ for ResNet-18). This advantage stems from VT pretraining, which enables learning robust feature representations. Thus, after $D_f$ examples are removed and the model is retrained, $\theta_r$ remains close to the original $\theta_o$ for VTs, explaining their high baseline "Original" performance (Table 11). However, this pretraining advantage diminishes on the more complex datasets (CIFAR-100).

Figure 1 (and Figure 5 in Appendix) shows how method rankings shift across architectures and proxies. With the Confidence proxy, Fine-tune and NegGrad+ perform best in ResNets, while VTs lag slightly. Under Holdout Retraining, however, VTs close or surpass this gap, particularly for NegGrad+. SalUn behaves differently: it achieves good ToW scores in VTs but underperforms on ToW-MIA compared to CNNs, suggesting that while it effectively adjusts transformer outputs to match $\theta_r$, it struggles to protect against MIAs.

**Key Takeaways.** Pretraining gives VTs an advantage on simpler tasks, but this weakens with increasing complexity. MU method rankings from CNNs do not transfer to VTs; they depend strongly on architectures and proxies. Holdout Retraining narrows or reverses the gap in favor of VTs.

### 4.3 How do VT architectures and capacities affect MU performance?

We observe systematic differences between ViT and Swin-T, as well as clear effects of model size.

**Architecture.** Swin-T exhibits stronger memorization than ViT (e.g., higher mean Feldman scores, heavier tails on CIFAR-10: $\mu$=0.17 vs. 0.09, see Figures 3 and 4), which likely explains its superior performance with gradient-based methods such as NegGrad+. Swin-T also aligns with CNN-like behavior in SalUn, achieving optimal performance at saliency threshold $\gamma = 0.3$, consistent with CNNs (Zhao et al., 2024), whereas ViT requires a much lower $\gamma = 0.1$. This suggests that ViT's global attention leads to more diffuse parameter involvement, while Swin-T's local windowed attention allows for more concentrated, targeted unlearning. In terms of algorithm performance, Figure 1 (and Figure 5, Tables 11 in Appendix) shows that Fine-tune is particularly effective on ViT (e.g., ToW-MIA=0.919 on CIFAR-10, 0.831 on CIFAR-100), while NegGrad+ excels on Swin-T, especially with Holdout Retraining on more complex tasks (e.g., ToW=0.975, ToW-MIA=0.902 on CIFAR-100). SalUn consistently attains good ToW but struggles on ToW-MIA, especially for ViT-Small (e.g., 0.582 on CIFAR-10 with Confidence). Dataset-level effects mirror these trends: ViT has an edge on the smaller, medium-complexity CIFAR-10, Swin-T dominates on the larger, more complex CIFAR-100, and both perform well on the simpler SVHN dataset. Continual unlearning results in Section 4.5 show broadly similar trends across ViT and Swin-T.

**Key Takeaways.** ViT favors fine-tuning-based unlearning, likely due to its global attention. Swin-T better supports gradient-based unlearning (NegGrad+) likely due to its local, hierarchical attention. Dataset complexity further shapes which family is most effective.

**Capacity.** To study the effect of model capacity, we extend our analysis to smaller and larger variants within each family, evaluated on CIFAR-10: ViT-Tiny ($\sim 5.5M$ params) vs. ViT-Small ($\sim 21.6M$) and Swin-Tiny ($\sim 27.5M$) vs. Swin-Small ($\sim 48.8M$). We focus on the Holdout Retraining (HR) proxy, which was especially promising in earlier results. We report ToW and ToW-MIA for the three MU algorithms (Fine-tune, NegGrad+, SalUn) alongside the Original baseline in Table 3. The headline findings continue to hold: Fine-tune and NegGrad+ remain consistently strong; SalUn attains high ToW but is often much weaker on ToW-MIA and is notably sensitive to model size, architecture, and proxy choice. We also observe *architecture-specific capacity trends:* For Swin-T, increasing capacity from Tiny to Small offers little ToW improvement and can reduce ToW-MIA, suggesting Swin-Tiny is already sufficient for CIFAR-10 and that Swin-Small may overfit. For

Table 2: $D_f$ accuracies of model $\theta_r$ across architectures.

| Architecture | CIFAR-10 | | CIFAR-100 | |
|---|---|---|---|---|
| | Confidence | Holdout Retraining | Confidence | Holdout Retraining |
| ResNet-18 | $50.433 \pm 6.808$ | $62.922 \pm 4.681$ | – | – |
| ResNet-50 | – | – | $64.267 \pm 0.504$ | $69.856 \pm 2.620$ |
| ViT-Small | $94.089 \pm 0.456$ | $89.244 \pm 1.321$ | $69.322 \pm 1.788$ | $68.767 \pm 3.828$ |
| Swin-Tiny | $91.867 \pm 1.711$ | $86.389 \pm 1.246$ | $69.833 \pm 1.242$ | $70.267 \pm 2.848$ |

ViT, the opposite holds: ViT-Tiny underperforms ViT-Small on both ToW and ToW-MIA, implying under-capacity limits MU effectiveness. The results reveal a "sweet spot" around ViT-Small and Swin-Tiny, where models are neither under- nor overfitting, yielding balanced unlearning-privacy.

**Key Takeaways.** Performance trends remain stable across ViT-Tiny/Small and Swin-Tiny/Small.

Table 3: ToW and ToW-MIA on CIFAR-10 with HR across four VT architectures.

| Algorithm | ViT-Small | ViT-Tiny | Swin-Small | Swin-Tiny |
|---|---|---|---|---|
| Original | $0.891 \pm 0.016$ | $0.773 \pm 0.020$ | $0.886 \pm 0.013$ | $0.862 \pm 0.008$ |
| Fine-tune | $0.928 \pm 0.013$ | $0.862 \pm 0.060$ | $0.921 \pm 0.005$ | $0.923 \pm 0.021$ |
| NegGrad+ | $0.916 \pm 0.016$ | $0.957 \pm 0.011$ | $0.944 \pm 0.028$ | $0.977 \pm 0.023$ |
| SalUn | $0.956 \pm 0.003$ | $0.846 \pm 0.043$ | $0.950 \pm 0.031$ | $0.961 \pm 0.048$ |

(a) ToW

| Algorithm | ViT-Small | ViT-Tiny | Swin-Small | Swin-Tiny |
|---|---|---|---|---|
| Original | $0.831 \pm 0.014$ | $0.699 \pm 0.020$ | $0.832 \pm 0.020$ | $0.811 \pm 0.018$ |
| Fine-tune | $0.913 \pm 0.008$ | $0.854 \pm 0.060$ | $0.924 \pm 0.010$ | $0.931 \pm 0.021$ |
| NegGrad+ | $0.968 \pm 0.015$ | $0.869 \pm 0.011$ | $0.888 \pm 0.021$ | $0.924 \pm 0.028$ |
| SalUn | $0.766 \pm 0.039$ | $0.640 \pm 0.043$ | $0.568 \pm 0.093$ | $0.834 \pm 0.022$ |

(b) ToW-MIA

## 4.4 Results on Larger/More Complex Data

To examine unlearning larger-scale, more complex conditions, we leverage that our Vision Transformers are pretrained on ImageNet-1K and use the *validation* split (50,000 images unseen during pretraining) as a new evaluation dataset (already $5\times$ larger than CIFAR-100's test set). We form a forget set of $|D_f|{=}3{,}000$ (partitioned into three 1k subsets for low/medium/high proxy values, as in Section 4.2), and treat the remaining 47k images as retain/test set. As this setup has only retain/test and forget components, we report ToW and ToW-MIA with two terms (retain/test accuracy and forget accuracy). We evaluate on the (more appropriate) larger Swin-Small (48.8M params) and use the Holdout Retraining proxy since it was previously found to be high performing.

Table 4: Performance with HR for Swin-Small on the 50k ImageNet-1K validation set.

| Algorithm | ToW | ToW-MIA |
|---|---|---|
| Fine-tune | $0.780 \pm 0.023$ | $0.747 \pm 0.009$ |
| NegGrad+ | $0.819 \pm 0.018$ | $0.772 \pm 0.034$ |
| SalUn | $0.743 \pm 0.027$ | $0.647 \pm 0.030$ |

Table 4 shows that the key earlier conclusions continue to hold at this larger scale: *NegGrad+* and *Fine-tune* perform strongly, while *SalUn* again underperforms, especially on ToW-MIA. This reinforces the conclusion that comparatively simple unlearning strategies can be effective even for larger models and datasets, albeit performance varies with architecture and data complexity. We also note that absolute ToW/ToW-MIA values are lower here than for Swin-Tiny on CIFAR-100, despite Swin-Small's larger ca-

Table 5: $D_f$ accuracies of Retrained $\theta_r$ (with HR) across architectures on the 50k ImageNet-1K validation set.

| Architecture | $D_f$ accuracy (%) |
|---|---|
| ResNet-50 | $69.139 \pm 1.722$ |
| Swin-Small | $69.433 \pm 2.246$ |

pacity. This is reasonable: (i) retain/test accuracy is harder given ImageNet's larger scale, diversity, and label space , and (ii) forgetting is more challenging on richer ImageNet images (greater embedding entanglement). Consequently, both terms in ToW and ToW-MIA are pressured downward relative to the CIFAR-100 setting, and further gains would likely require even larger-capacity models and/or stronger regularization. Finally, Table 5 reinforces our earlier observation that the benefit of pretraining for forget accuracy diminishes as task complexity increases (see Table 2).

**Key Takeaways.** Simple methods (NegGrad+, Fine-tune) remain effective even on ImageNet-scale data, whereas SalUn underperforms, and the benefits of pretraining diminish as complexity grows.

## 4.5 Does Continual Unlearning Impact Performance in Vision Transformers?

We selected NegGrad+ and Holdout Retraining, being the best-performing algorithm-proxy pair, as shown earlier. We applied five sequential unlearning steps, unlearning the same total number of

examples (3,000) as before. Proxy values are recalculated after each unlearning step. We use three partitions as before, but set $|D_f| = 600$ per step, resulting in $N = 200$ examples per partition.

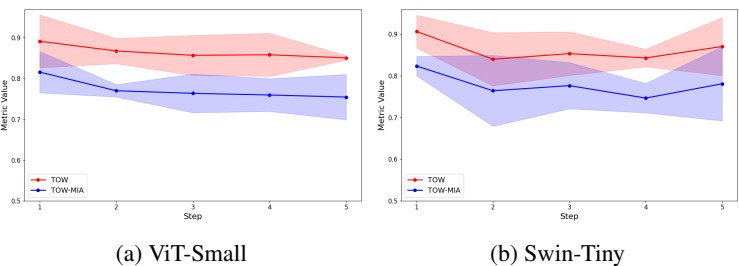

(a) ViT-Small  (b) Swin-Tiny

Figure 2: Continual Unlearning Performance Across Five Steps on CIFAR-100

Figure 2 (and Table 14, Appendix A.3.2 for CIFAR-10 and Appendix A.7.2 for SVHN) show the unlearning performance across five consecutive steps, with 95% confidence intervals. It shows stability across metrics, VTs and datasets. ViT-Small suffers a slight downward trend. But even after five unlearning steps, all metrics remain within the 95% confidence intervals of the initial unlearning step and similar to those in Section 4.2.

**Key Takeaway.** Minimal (if any) degradation occurs under continual unlearning in VTs.

## 5 LIMITATIONS

Our benchmarking setup represents a practical compromise in terms of VT models, their sizes, and the size and complexity of datasets, allowing for insightful comparisons to CNNs, while studying fundamental approaches to MU. Nonetheless, there is much more work to follow, adding more datasets, more models and capacities, more (new and existing) algorithms, and different metrics.

## 6 CONCLUSIONS

We benchmarked the performance of fundamentally different approaches to MU, ranging from simpler baselines to more advanced SOTA unlearning algorithms in VTs. We focused on algorithms leveraging memorization, as memorization has been found to be key to unlearning (across modalities) and the SOTA MU approaches for vision tasks can be substantially improved by leveraging it. To ensure the practicality of leveraging memorization, we studied memorization patterns in VTs and five different memorization proxies that have been found to offer high memorization-score fidelity in CNNs while being drastically more efficient to compute. We employed different datasets to account for the effects of varying sizes and complexities. We studied two different popular VT architectures (ViT and Swin-T) selected to be closer or further from CNN inductive biases. We also used different model capacities to see the impact of capacities on MU performance (with carefully selected dataset sizes to avoid over- and under-fitting). This work represents the first comprehensive study in this domain and sheds new light on the current state of affairs of MU algorithm performance in VTs.

This work contributes the following novel insights: (i) Pretraining of VTs can help improve unlearning for smaller and simpler datasets/tasks. (ii) VTs and CNNs largely exhibit the same memorization patterns. (iii) Well-known memorization proxies from CNNs can benefit unlearning performance in VTs as they can do in CNNs. (iv) Continual unlearning does not degrade the efficacy of memorization proxies and overall unlearning performance in VTs. (v) VTs can enjoy similar, if not better, unlearning performance to that in CNNs (using CNN-derived algorithms). (vi) Swin-T can outperform ViT on more complex datasets, due to Swin-T's architectural similarities to CNNs. (vii) NegGrad+, perhaps surprisingly, emerges as a consistent, strong performer for VTs, especially on more complex datasets. (viii) Holdout Retraining emerges as the proxy yielding superior unlearning performance in VTs, whereas Confidence can be competitive on lower-complexity datasets. We hope these insights, as well as the associated publicly available codebase/dataset infrastructure benchmark, will springboard the study of MU in VTs, serving as a basis upon which to add new (future) algorithms, especially developed for VTs.

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

# A  TECHNICAL APPENDICES AND SUPPLEMENTARY MATERIAL

CONTENTS

## A.1  IMPLEMENTATION DETAILS

This section outlines the key implementation details for all experiments conducted in this study. For all experiments involving transformer architectures, we fine-tune models initialized with ImageNet-1k pretrained weights provided by the `timm` library Wightman (2019), as described in Section 3.2.

All experiments on the CIFAR-10 and CIFAR-100 datasets were conducted using a mix of NVIDIA RTX 2080 Ti, NVIDIA A10 and NVIDIA TITAN Xp GPUs, taking roughly 210 GPU hours. For the experiments on the SVHN dataset, training and evaluation were performed on NVIDIA A5000 GPUs, which consumed approximately 300 GPU hours.

The code for reproducing the results is available at: `https://anonymous.4open.science/r/unlearning_VTs-31E1`

## A.1.1  MEMORIZATION ESTIMATION

For memorization estimation, we use ResNet-18 for CIFAR-10 and rely on precomputed memorization scores provided by Feldman and Zhang (2020) for CIFAR-100. Additionally, we compute memorization scores using ViT-Small and Swin-Tiny architectures for both CIFAR-10 and CIFAR-100. The training configurations and corresponding hyperparameters are summarized in Table 6.

Table 6: Training Configurations along with hyper-parameters of the models used to compute memorization scores shown in Figures 3 and 4. The "Transformers" column represents the configuration for both ViT-Small and Swin-Tiny across CIFAR-10 and CIFAR-100.

|  | ResNet-18 | Transformers |
| --- | --- | --- |
| **Optimizer** | SGD | AdamW |
| **Base learning rate** | 0.01 | 0.0001 |
| **Loss** | Cross-Entropy | Cross-Entropy |
| **Learning rate scheduler** | Step decay | CosineAnnealingLR |
| **Batch size** | 512 | 128 |
| **Epochs** | 30 | 30 |
| **Momentum** | 0.9 | - |
| **Weight decay** | 0.0005 | 0.05 |
| **Data augmentation** | None | None |

### A.1.2 UNLEARNING

This section presents the configuration and hyper-parameter details used to obtain the necessary models for ToW and ToW-MIA: "Original" $\theta_o$, "Retrained" $\theta_r$ and "Unlearned" $\theta_u$. Table 7 shows the hyper-parameter details for training original models & retraining. Table 8 shows the unlearning hyper-parameters, and Table 9 presents the sequential unlearning hyper-parameters.

Table 7: Training configurations and hyperparameter used for the original models ($\theta_o$) and the retrained models ($\theta_r$) on the CIFAR-10 and CIFAR-100 datasets.

| Hyperparameter | ViT-Small & Swin-Tiny |
| --- | --- |
| **Optimizer** | AdamW |
| **Base learning rate** | 0.0001 |
| **Loss** | Cross-Entropy |
| **Learning rate scheduler** | CosineAnnealingLR |
| **Batch size** | 128 |
| **Epochs** | 50 |
| **Weight decay** | 0.05 |
| **Data augmentation** | Random Crop + Horizontal Flip |

Table 8: Hyper-parameters across all unlearn methods within the RUM$_F$ meta-algorithm on both Confidence and Holdout Retraining proxies to derive the unlearn models $\theta_u$. "Unlearn Epochs" column refers to the performed number of epochs each algorithm performed for the three memorization partitions (low $\rightarrow$ medium $\rightarrow$ high) respectively.

| Algorithm | Architecture | Unlearn Epochs | Unlearn LR | $\beta$ | $\alpha$ | $\gamma$ |
| --- | --- | --- | --- | --- | --- | --- |
| FineTune | ViT-Small | 5,5,10 | 0.0001 | – | – | – |
|  | Swin-Tiny | 5,5,10 | 0.0001 | – | – | – |
| NegGrad+ | ViT-Small | 5,5,10 | 0.00002 | 0.97 | – | – |
|  | Swin-Tiny | 5,5,10 | 0.00002 | 0.97 | – | – |
| SalUn | ViT-Small | 5,5,10 | 0.00005 | – | 1 | 0.1 |
|  | Swin-Tiny | 5,5,10 | 0.0002 | – | 1 | 0.3 |

Table 9: Hyper-parameters at each sequential unlearning step for the NegGrad+ and Holdout Retraining configuration utilized. "Unlearn Epochs" are reduced to 1 for partition (low → medium → high) due to the smaller forget set size($|D_f| = 600$) at each step. All other settings remain consistent with the primary experiments.

| Unlearn Epochs | Unlearn LR | $\beta$ |
|:---:|:---:|:---:|
| 1,1,1 | 0.00002 | 0.97 |

## A.2 MEMORIZATION AND PROXIES IN VTS

### A.2.1 DO VISION TRANSFORMERS MEMORIZE THE SAME WAY AS CNNS?

As memorization plays a key role in unlearning, this step is critical. For ResNet-18/CIFAR-10 and both VTs for both datasets, the training configuration is detailed in Table 6 (Appendix A.1.1). For ResNet-50/CIFAR-100, scores were already pre-computed Feldman and Zhang (2020).

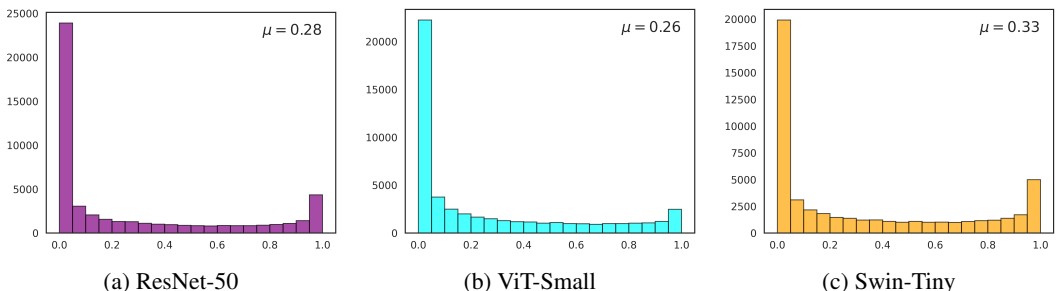

(a) ResNet-50      (b) ViT-Small      (c) Swin-Tiny

Figure 3: Memorization histograms across different architectures on CIFAR-100

Figures 3 (and 4 present the memorization distributions across different architectures for CIFAR-100 (CIFAR-10). In all cases we see similarly skewed distributions, consistent with the "long-tail" discovery in Feldman and Zhang (2020). For CIFAR-100, in particular, both VTs closely align with ResNet-50. Hence, despite architectural differences between VTs and CNNs, their memorization patterns remain fundamentally similar, especially for more complex tasks. This provides a foundation for studying unlearning a la RUM in VTs.

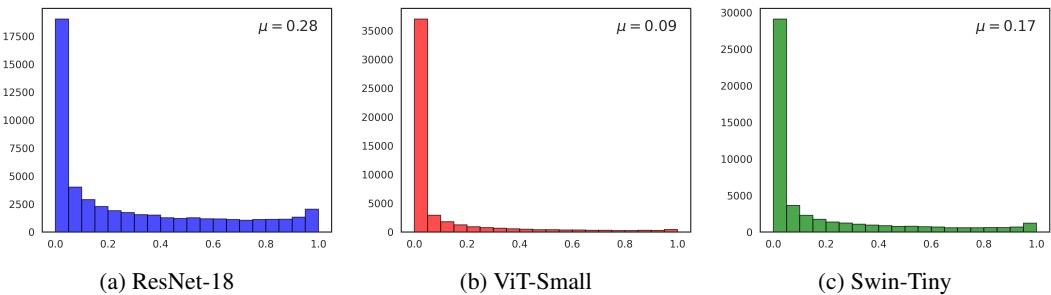

(a) ResNet-18      (b) ViT-Small      (c) Swin-Tiny

Figure 4: Memorization histograms across different architectures on CIFAR-10

For CIFAR-100, we observe that both VTs closely align with ResNet-50 in terms of memorization. However, on CIFAR-10, VTs exhibit slightly lower memorization compared to ResNet-18. This can be attributed to the fact that pre-trained VTs benefit from a stronger ability to capture global contextual information, which reduces their reliance on memorizing training examples when adapting to simpler tasks. This advantage, however, does not extend to more challenging tasks like CIFAR-100.

A.2.2   ARE CNN-DERIVED MEMORIZATION PROXIES RELEVANT FOR VTS?

Differences in architecture (e.g., inductive biases) and in training (e.g., finetune-then-retrain in VTs) raise doubts about the appropriateness of CNN memorization proxies for VTs.

We first provide the formulas for computing each of the following proxies: Confidence (C), Max Confidence (MaxC), Entropy (E), Binary Accuracy (BA) Jiang et al. (2021) and Holdout Retraining (HR) Carlini et al. (2019).

$$C(x_i, y_i) = \frac{1}{E} \sum_{e=1}^{E} P_{\theta_e}(y = y_i \mid x_i), \tag{7a}$$

$$\text{MaxC}(x_i) = \frac{1}{E} \sum_{e=1}^{E} \max_{y} P_{\theta_e}(y \mid x_i), \tag{7b}$$

$$E(x_i) = \frac{1}{E} \sum_{e=1}^{E} \sum_{y} P_{\theta_e}(y \mid x_i) \log P_{\theta_e}(y \mid x_i), \tag{7c}$$

$$\text{BA}(x_i, y_i) = \frac{1}{E} \sum_{e=1}^{E} \mathbb{1} \left[ \arg \max_{y} P_{\theta_e}(y \mid x_i) = y_i \right], \tag{7d}$$

$$\text{HR}(x_i) = \text{symKL}\big(p_{\theta_o}(x_i) \,\|\, p_{\theta'}(x_i)\big). \tag{7e}$$

Results in Table 1 answer the question in the affirmative. Proxies were calculated using models trained with identical hyper-parameters as those used to derive memorization scores (see Table 6 in Appendix A.1.1).

Table 10: Spearman Correlation Coefficients Between Memorization and Proxies (duplicate of Table 1, included here for completeness)

| Proxy | CIFAR-10 | | | CIFAR-100 | | |
|---|---|---|---|---|---|---|
| | ResNet-18 | ViT-Small | Swin-Tiny | ResNet-50 | ViT-Small | Swin-Tiny |
| **Confidence** | -0.80 | -0.79 | -0.88 | -0.91 | -0.85 | -0.90 |
| Max Confidence | -0.76 | -0.77 | -0.85 | -0.87 | -0.80 | -0.86 |
| Entropy | -0.75 | -0.78 | -0.85 | -0.80 | -0.77 | -0.82 |
| Binary Accuracy | -0.71 | -0.63 | -0.79 | -0.89 | -0.69 | -0.78 |
| **Holdout Retraining** | +0.67 | +0.45 | +0.64 | +0.62 | +0.50 | +0.52 |

Confidence consistently demonstrates the strongest correlation across all models and datasets. The correlation magnitude for VTs largely resembles that of CNNs – further evidence that memorizations are fundamentally similar across these architectures. Swin-Tiny generally has stronger correlations compared to ViT-Small, possibly due to its hierarchical structure that more closely resembles traditional CNNs. Holdout Retraining shows more moderate positive (albeit still significant) correlations.

Based on these findings, we selected Confidence as the best-performing learning event proxy with consistently high absolute correlation values and Holdout Retraining as a representative of a different proxy category. Importantly, Holdout Retraining offers large computational advantages as it does not require monitoring model behaviour during training.

A.3   RESULTS FOR CIFAR-10

This section presents the corresponding figures for the CIFAR-10 dataset that mirror the ones of CIFAR-100 found in Section 4.

A.3.1   UNLEARNING PERFORMANCE FOR CIFAR-10

**Algorithm-Level Comparison on CIFAR-10.** Figure 5 shows the unlearning performance results for the CIFAR-10 dataset. First, note that the original VT model (no unlearning) already achieves

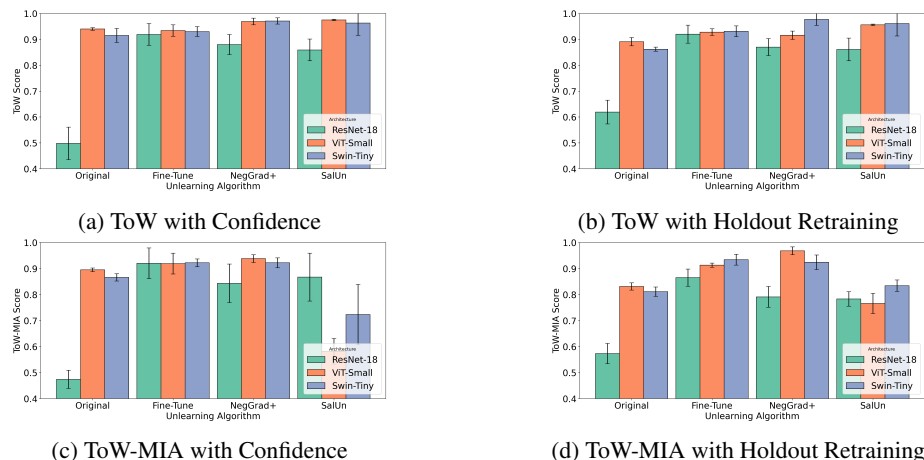

(a) ToW with Confidence

(b) ToW with Holdout Retraining

(c) ToW-MIA with Confidence

(d) ToW-MIA with Holdout Retraining

Figure 5: Architecture performance comparison on CIFAR-10

relatively high scores on CIFAR-10, which is unlike the results in CIFAR-100. This is consistent with our earlier observation that VTs exhibit lower memorization on this simpler dataset, reducing performance differences between $\theta_o$ and $\theta_r$ on $D_f$. Nevertheless, we see marginal improvements from unlearning algorithms on CIFAR-10, still making a case for their utilization in low-memorized data.

Taking a closer look, all methods in VTs appear to show smaller improvements in ToW compared to the original model. Additionally, there is no clear winner for VTs between the two proxy strategies. Again, these results are in contrast to the more substantial improvements observed on the more complex CIFAR-100 dataset. For instance, even a simple approach like Fine-tune significantly outperforms the original model across all configurations on CIFAR-100—an effect not observed here. More advanced algorithms show greater gains in ToW performance on CIFAR-100 than they do on CIFAR-10, compared to the original VT model.

**Architecture-Level Comparison on CIFAR-10.** Figure 5 also presents the architecture-wise comparison on CIFAR-10, analogous to the analysis shown for CIFAR-100 in Figure 1. We observe that VTs generally achieve higher ToW scores than ResNet-18, with NegGrad+ and SalUn performing particularly well on both VTs. However, this performance gap is less evident in the ToW-MIA metrics, where Fine-tune shows comparable performance across all architectures. Notably, NegGrad+ is the only unlearning method that improves ToW-MIA for both proxies. In contrast, SalUn continues to struggle on ToW-MIA within VTs: ResNet-18 significantly outperforms VTs when using the Confidence proxy and matches their performance under Holdout Retraining.

### A.3.2 CONTINUAL UNLEARNING FOR CIFAR-10

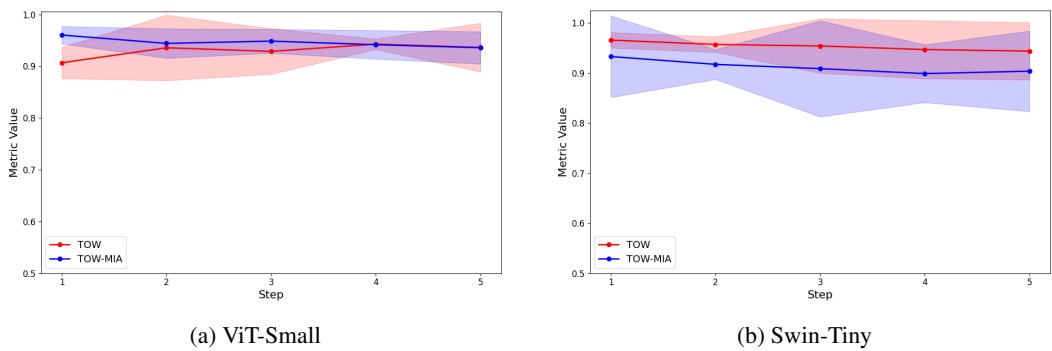

(a) ViT-Small

(b) Swin-Tiny

Figure 6: Continual Unlearning Performance Across Multiple Steps on CIFAR-10

Table 11: Detailed performance results for the two VTs across both the CIFAR datasets.

| Algorithm | Confidence | | Holdout Ret | |
|---|---|---|---|---|
| | ToW ($\uparrow$) | ToW-MIA ($\uparrow$) | ToW ($\uparrow$) | ToW-MIA ($\uparrow$) |
| Original | $0.940 \pm 0.005$ | $0.895 \pm 0.007$ | $0.891 \pm 0.016$ | $0.831 \pm 0.014$ |
| Fine-tune | $0.934 \pm 0.022$ | $0.919 \pm 0.040$ | $0.928 \pm 0.013$ | $0.913 \pm 0.008$ |
| NegGrad+ | $0.969 \pm 0.013$ | $0.938 \pm 0.015$ | $0.916 \pm 0.016$ | $0.968 \pm 0.015$ |
| SalUn | $0.975 \pm 0.003$ | $0.582 \pm 0.049$ | $0.956 \pm 0.003$ | $0.766 \pm 0.039$ |

(a) CIFAR-10 – ViT-Small

| Algorithm | Confidence | | Holdout Ret | |
|---|---|---|---|---|
| | ToW ($\uparrow$) | ToW-MIA ($\uparrow$) | ToW ($\uparrow$) | ToW-MIA ($\uparrow$) |
| Original | $0.915 \pm 0.027$ | $0.866 \pm 0.014$ | $0.862 \pm 0.008$ | $0.811 \pm 0.018$ |
| Fine-tune | $0.930 \pm 0.019$ | $0.922 \pm 0.015$ | $0.923 \pm 0.021$ | $0.931 \pm 0.021$ |
| NegGrad+ | $0.971 \pm 0.012$ | $0.922 \pm 0.019$ | $0.977 \pm 0.023$ | $0.924 \pm 0.028$ |
| SalUn | $0.963 \pm 0.048$ | $0.723 \pm 0.116$ | $0.961 \pm 0.048$ | $0.834 \pm 0.022$ |

(b) CIFAR-10 – Swin-Tiny

| Algorithm | Confidence | | Holdout Ret | |
|---|---|---|---|---|
| | ToW ($\uparrow$) | ToW-MIA ($\uparrow$) | ToW ($\uparrow$) | ToW-MIA ($\uparrow$) |
| Original | $0.629 \pm 0.038$ | $0.521 \pm 0.026$ | $0.619 \pm 0.040$ | $0.538 \pm 0.023$ |
| Fine-tune | $0.813 \pm 0.008$ | $0.831 \pm 0.014$ | $0.855 \pm 0.036$ | $0.889 \pm 0.008$ |
| NegGrad+ | $0.844 \pm 0.034$ | $0.736 \pm 0.023$ | $0.931 \pm 0.039$ | $0.838 \pm 0.024$ |
| SalUn | $0.923 \pm 0.019$ | $0.602 \pm 0.014$ | $0.903 \pm 0.049$ | $0.709 \pm 0.085$ |

(c) CIFAR-100 – ViT-Small

| Algorithm | Confidence | | Holdout Ret | |
|---|---|---|---|---|
| | ToW ($\uparrow$) | ToW-MIA ($\uparrow$) | ToW ($\uparrow$) | ToW-MIA ($\uparrow$) |
| Original | $0.665 \pm 0.014$ | $0.553 \pm 0.015$ | $0.670 \pm 0.027$ | $0.586 \pm 0.025$ |
| Fine-tune | $0.774 \pm 0.030$ | $0.758 \pm 0.015$ | $0.822 \pm 0.034$ | $0.839 \pm 0.032$ |
| NegGrad+ | $0.867 \pm 0.021$ | $0.767 \pm 0.019$ | $0.975 \pm 0.027$ | $0.902 \pm 0.035$ |
| SalUn | $0.835 \pm 0.060$ | $0.557 \pm 0.034$ | $0.870 \pm 0.039$ | $0.751 \pm 0.043$ |

(d) CIFAR-100 – Swin-Tiny

Figure 6 an Table 14 show the results in the same fashion as seen in Figure 2 for CIFAR-100. Here we observe an even higher stability in results with even more negligible degradation, further accentuated by the smaller performance gap between ToW and ToW-MIA to those seen for CIFAR-100. This along with higher average scores comes to no surprise due to the higher "Original" baseline performance exhibited by the VTs on CIFAR-10.

## A.4 DETAILED ToW AND ToW-MIA RESULTS ON CIFAR-10 AND CIFAR-100

This section presents detailed ToW and ToW-MIA performance results for all evaluated architectures on the CIFAR-10 and CIFAR-100 datasets. These results for ViT-Small, Swin-Tiny (Table 11), and ResNets (Table 12) are used to generate Figures 1 and 5.

### A.4.1 RESULTS FOR ViT-SMALL AND SWIN-TINY ACROSS ALL ALGORITHMS

Table 11 provides detailed ToW and ToW-MIA results for ViT-Small and Swin-Tiny on CIFAR-10/CIFAR-100. These values are used in the corresponding architecture and method comparison figures.

### A.4.2 RESULTS FOR RESNET-18 AND RESNET-50 ACROSS ALL ALGORITHMS

Table 12 reports the detailed results for ResNet-18 (CIFAR-10) and ResNet-50 (CIFAR-100). These metrics support the comparative analysis shown in Figures 1 and 5.

Table 12: Detailed performance result metrics for ResNet-18 and ResNet-50 on the CIFAR-10 and CIFAR-100 datasets respectively.

| | Confidence | | Holdout Retraining | |
|---|---|---|---|---|
| **Algorithm** | **ToW** ($\uparrow$) | **ToW-MIA** ($\uparrow$) | **ToW** ($\uparrow$) | **ToW-MIA** ($\uparrow$) |
| Fine-tune | $0.919 \pm 0.042$ | $0.920 \pm 0.059$ | $0.920 \pm 0.035$ | $0.865 \pm 0.033$ |
| NegGrad+ | $0.880 \pm 0.039$ | $0.843 \pm 0.074$ | $0.870 \pm 0.033$ | $0.791 \pm 0.040$ |
| SalUn | $0.859 \pm 0.042$ | $0.867 \pm 0.092$ | $0.861 \pm 0.044$ | $0.783 \pm 0.028$ |

(a) CIFAR-10 with ResNet-18

| | Confidence | | Holdout Retraining | |
|---|---|---|---|---|
| **Algorithm** | **ToW** ($\uparrow$) | **ToW-MIA** ($\uparrow$) | **ToW** ($\uparrow$) | **ToW-MIA** ($\uparrow$) |
| Fine-tune | $0.863 \pm 0.049$ | $0.857 \pm 0.059$ | $0.846 \pm 0.032$ | $0.792 \pm 0.044$ |
| NegGrad+ | $0.890 \pm 0.047$ | $0.922 \pm 0.017$ | $0.866 \pm 0.031$ | $0.838 \pm 0.027$ |
| SalUn | $0.665 \pm 0.031$ | $0.636 \pm 0.038$ | $0.696 \pm 0.032$ | $0.699 \pm 0.034$ |

(b) CIFAR-100 with ResNet-50

### A.5 DETAILED ACCURACY RESULTS FOR VTS ARCHITECTURES ACROSS DATASETS

Table 13: Accuracies on $D_r$, $D_f$ and $D_{test}$ for the "Retrain" models $\theta_r$ as well as the respective "Unlearned" models $\theta_u$, evaluated across all unlearning algorithms within the RUM$_F$ framework. MIA performance is also reported. Results are averaged over 3 runs, with 95% confidence intervals.

| | Confidence | | | | Holdout Retaining | | | |
|---|---|---|---|---|---|---|---|---|
| **Algorithm** | **Retain Acc** | **Forget Acc** | **Test Acc** | **MIA** | **Retain Acc** | **Forget Acc** | **Test Acc** | **MIA** |
| Retrain | $99.969 \pm 0.006$ | $94.089 \pm 0.456$ | $93.703 \pm 0.486$ | $0.108 \pm 0.007$ | $99.974 \pm 0.035$ | $89.244 \pm 1.321$ | $93.767 \pm 0.029$ | $0.174 \pm 0.012$ |
| Fine-tune | $99.433 \pm 0.639$ | $94.111 \pm 0.912$ | $87.947 \pm 1.519$ | $0.128 \pm 0.014$ | $99.562 \pm 0.347$ | $87.889 \pm 2.356$ | $88.240 \pm 0.348$ | $0.204 \pm 0.016$ |
| NegGrad+ | $99.832 \pm 0.140$ | $93.411 \pm 1.078$ | $91.407 \pm 0.872$ | $0.069 \pm 0.008$ | $99.814 \pm 0.073$ | $83.289 \pm 2.115$ | $91.340 \pm 0.413$ | $0.167 \pm 0.020$ |
| SalUn | $100.000 \pm 0.000$ | $96.033 \pm 0.543$ | $93.203 \pm 0.617$ | $0.523 \pm 0.057$ | $100.000 \pm 0.000$ | $93.089 \pm 1.971$ | $93.180 \pm 0.523$ | $0.403 \pm 0.042$ |

(a) CIFAR-10 with ViT-Small

| | Confidence | | | | Holdout Retaining | | | |
|---|---|---|---|---|---|---|---|---|
| **Algorithm** | **Retain Acc** | **Forget Acc** | **Test Acc** | **MIA** | **Retain Acc** | **Forget Acc** | **Test Acc** | **MIA** |
| Retrain | $99.961 \pm 0.021$ | $91.867 \pm 1.711$ | $91.607 \pm 0.596$ | $0.133 \pm 0.002$ | $99.979 \pm 0.027$ | $86.389 \pm 1.246$ | $91.767 \pm 0.689$ | $0.190 \pm 0.020$ |
| Fine-tune | $99.833 \pm 0.141$ | $96.733 \pm 0.840$ | $89.463 \pm 0.660$ | $0.076 \pm 0.005$ | $99.779 \pm 0.012$ | $91.478 \pm 0.208$ | $89.183 \pm 0.352$ | $0.148 \pm 0.009$ |
| NegGrad+ | $99.968 \pm 0.018$ | $93.833 \pm 0.597$ | $90.643 \pm 1.096$ | $0.064 \pm 0.005$ | $99.903 \pm 0.024$ | $81.911 \pm 0.694$ | $90.473 \pm 0.254$ | $0.181 \pm 0.007$ |
| SalUn | $100.000 \pm 0.000$ | $93.967 \pm 8.103$ | $90.743 \pm 0.547$ | $0.403 \pm 0.126$ | $100.000 \pm 0.000$ | $88.533 \pm 9.399$ | $91.160 \pm 0.692$ | $0.351 \pm 0.043$ |

(b) CIFAR-10 with Swin-Tiny

| | Confidence | | | | Holdout Retaining | | | |
|---|---|---|---|---|---|---|---|---|
| **Algorithm** | **Retain Acc** | **Forget Acc** | **Test Acc** | **MIA** | **Retain Acc** | **Forget Acc** | **Test Acc** | **MIA** |
| Retrain | $98.457 \pm 0.819$ | $69.322 \pm 1.788$ | $69.670 \pm 2.445$ | $0.432 \pm 0.012$ | $98.063 \pm 0.413$ | $68.767 \pm 3.828$ | $69.467 \pm 1.032$ | $0.412 \pm 0.024.$ |
| Fine-tune | $99.325 \pm 0.148$ | $86.056 \pm 1.907$ | $68.193 \pm 1.578$ | $0.283 \pm 0.014$ | $99.322 \pm 0.622$ | $81.200 \pm 2.766$ | $68.350 \pm 1.433$ | $0.322 \pm 0.025$ |
| NegGrad+ | $99.846 \pm 0.166$ | $82.200 \pm 0.299$ | $71.440 \pm 1.793$ | $0.192 \pm 0.004$ | $99.637 \pm 0.177$ | $72.722 \pm 0.621$ | $70.997 \pm 0.448$ | $0.276 \pm 0.006$ |
| SalUn | $99.982 \pm 0.007$ | $64.756 \pm 0.669$ | $71.467 \pm 0.816$ | $0.810 \pm 0.014$ | $99.981 \pm 0.006$ | $63.878 \pm 2.161$ | $72.620 \pm 0.480$ | $0.666 \pm 0.068$ |

(c) CIFAR-100 with ViT-Small

| | Confidence | | | | Holdout Retaining | | | |
|---|---|---|---|---|---|---|---|---|
| **Algorithm** | **Retain Acc** | **Forget Acc** | **Test Acc** | **MIA** | **Retain Acc** | **Forget Acc** | **Test Acc** | **MIA** |
| Retrain | $99.220 \pm 0.167$ | $69.833 \pm 1.242$ | $68.923 \pm 0.296$ | $0.430 \pm 0.015$ | $99.143 \pm 0.185$ | $70.267 \pm 2.848$ | $69.057 \pm 0.827$ | $0.392 \pm 0.024$ |
| Fine-tune | $99.638 \pm 0.184$ | $91.811 \pm 3.019$ | $68.503 \pm 0.405$ | $0.194 \pm 0.022$ | $99.368 \pm 0.557$ | $87.289 \pm 1.300$ | $68.377 \pm 1.536$ | $0.239 \pm 0.016$ |
| NegGrad+ | $99.940 \pm 0.047$ | $81.900 \pm 0.955$ | $69.410 \pm 1.083$ | $0.207 \pm 0.002$ | $99.914 \pm 0.047$ | $70.111 \pm 0.723$ | $69.833 \pm 0.521$ | $0.308 \pm 0.009$ |
| SalUn | $99.965 \pm 0.003$ | $55.578 \pm 6.181$ | $70.750 \pm 0.676$ | $0.859 \pm 0.054$ | $99.966 \pm 0.007$ | $60.156 \pm 1.615$ | $71.490 \pm 0.696$ | $0.616 \pm 0.021$ |

(d) CIFAR-100 with Swin-Tiny

## A.6 CONTINUAL UNLEARNING RESULTS IN VTS FOR CIFAR-10, CIFAR-100

Table 14 presents detailed ToW and ToW-MIA results of VTs architectures over CIFAR-10 and CIFAR-100, at each sequential unlearning step using the strong-performing combination of the Neg-Grad+ algorithm with the Holdout Retraining proxy.

Table 14: Detailed unlearning performance at each of the 5 sequential steps for the NegGrad+ and Holdout Retraining configuration discussed in Section 4.5 and displayed in Figures 2 and 6.

| | CIFAR-10 | | | | CIFAR-100 | | | |
| | ViT-Small | | Swin-Tiny | | ViT-Small | | Swin-Tiny | |
| Step | ToW ($\uparrow$) | ToW-MIA ($\uparrow$) | ToW ($\uparrow$) | ToW-MIA ($\uparrow$) | ToW ($\uparrow$) | ToW-MIA ($\uparrow$) | ToW ($\uparrow$) | ToW-MIA ($\uparrow$) |
|---|---|---|---|---|---|---|---|---|
| 1 | $0.907 \pm 0.030$ | $0.960 \pm 0.017$ | $0.966 \pm 0.016$ | $0.933 \pm 0.082$ | $0.891 \pm 0.065$ | $0.816 \pm 0.051$ | $0.906 \pm 0.039$ | $0.823 \pm 0.023$ |
| 2 | $0.936 \pm 0.063$ | $0.944 \pm 0.029$ | $0.957 \pm 0.016$ | $0.917 \pm 0.030$ | $0.868 \pm 0.031$ | $0.770 \pm 0.015$ | $0.840 \pm 0.064$ | $0.764 \pm 0.085$ |
| 3 | $0.929 \pm 0.044$ | $0.948 \pm 0.023$ | $0.954 \pm 0.054$ | $0.909 \pm 0.096$ | $0.857 \pm 0.049$ | $0.764 \pm 0.047$ | $0.853 \pm 0.052$ | $0.776 \pm 0.055$ |
| 4 | $0.943 \pm 0.010$ | $0.942 \pm 0.028$ | $0.947 \pm 0.058$ | $0.899 \pm 0.058$ | $0.858 \pm 0.053$ | $0.760 \pm 0.040$ | $0.843 \pm 0.021$ | $0.746 \pm 0.035$ |
| 5 | $0.936 \pm 0.047$ | $0.936 \pm 0.031$ | $0.944 \pm 0.057$ | $0.904 \pm 0.080$ | $0.851 \pm 0.006$ | $0.754 \pm 0.056$ | $0.870 \pm 0.070$ | $0.781 \pm 0.089$ |

## A.7 RESULTS FOR SVHN

### A.7.1 DETAILED TOW AND TOW-MIA RESULTS ACROSS VTS FOR SVHN

To extend our study to a different dataset, we repeat the experiments on SVHN using both ViT-Small and Swin-Tiny architectures. Table 15 lists the hyperparameter settings used for the original model training and subsequent retraining.

Table 15: Training configurations and hyperparameters used for original models $\theta_o$ and retrained models $\theta_r$ alike, using ViT-Small and Swin-Tiny architectures on the SVHN dataset.

| Hyperparameter | ViT-Small | Swin-Tiny |
|---|---|---|
| **Optimizer** | AdamW | AdamW |
| **Base learning rate** | 0.0001 | 0.0001 |
| **Loss** | Cross-Entropy | Cross-Entropy |
| **Learning rate scheduler** | CosineAnnealingLR | CosineAnnealingLR |
| **Batch size** | 256 | 256 |
| **Epochs** | 30 | 30 |
| **Weight decay** | 0.01 | 0.05 |
| **Data augmentation** | - | - |

Following the procedure in Section 4.2, we apply all unlearn methods within the RUM$_F$ meta-algorithm to the SVHN dataset. We evaluate both ViT-Small and Swin-Tiny architectures, using Confidence and Holdout Retraining as proxy metrics. Figure 7 shows the comparative unlearning performance of these methods on SVHN. See Section 4.2 for a detailed discussion of the results.

### A.7.2 CONTINUAL UNLEARNING RESULTS IN VTS FOR SVHN

We follow the same procedure outlined in Section 4.5 to sequentially apply NegGrad+ and Holdout Retraining on the SVHN dataset over five consecutive steps. The results are presented in Figure 8, evaluated by ToW and ToW-MIA. They further support the observation from Section 4.5: continual unlearning in VTs exhibits high stability across steps, with minimal degradation in performance.

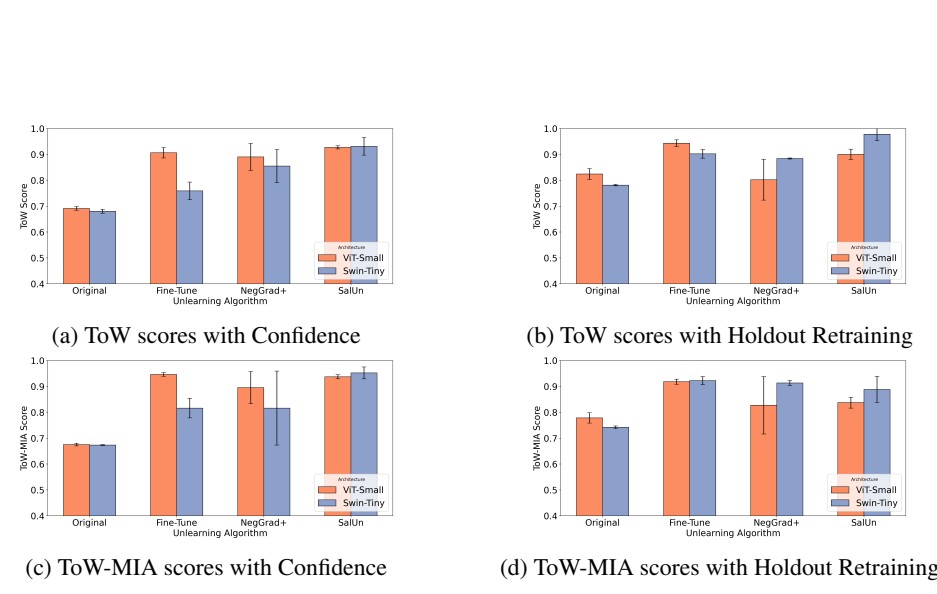

(a) ToW scores with Confidence

(b) ToW scores with Holdout Retraining

(c) ToW-MIA scores with Confidence

(d) ToW-MIA scores with Holdout Retraining

Figure 7: MU performance comparison on SVHN

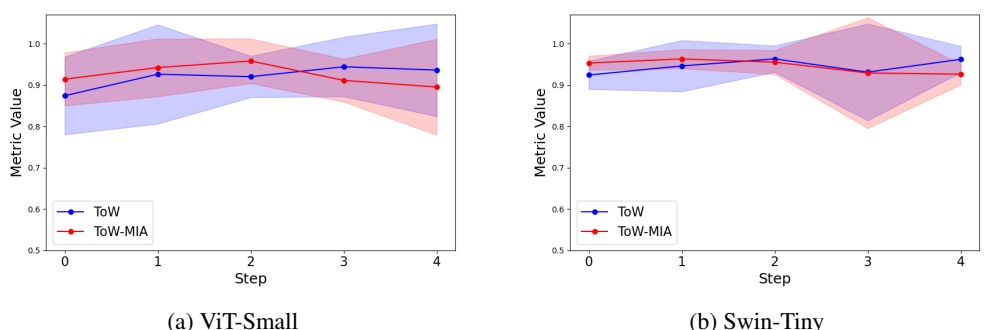

(a) ViT-Small

(b) Swin-Tiny

Figure 8: Continual Unlearning Performance Across Five Steps on SVHN

