## Qualitative Visualization of Forget vs. Retain Behavior

To complement our quantitative results, we include qualitative visualizations based on 2-D PCA projections of the models' embedding space. For each architecture (ViT-Small and Swin-Tiny), and for the same set of samples, we extract image-level embeddings from:

- the original (pre-unlearning) model, and
- the unlearned model (NegGrad+ with the HO-Ret proxy),

and then project both sets of embeddings into two dimensions using PCA.

We visualize four example classes. Within each class, retain examples are shown as dots, and forget examples are shown as crosses of the same color. The left panel of each figure shows the original model, and the right panel shows the unlearned model.

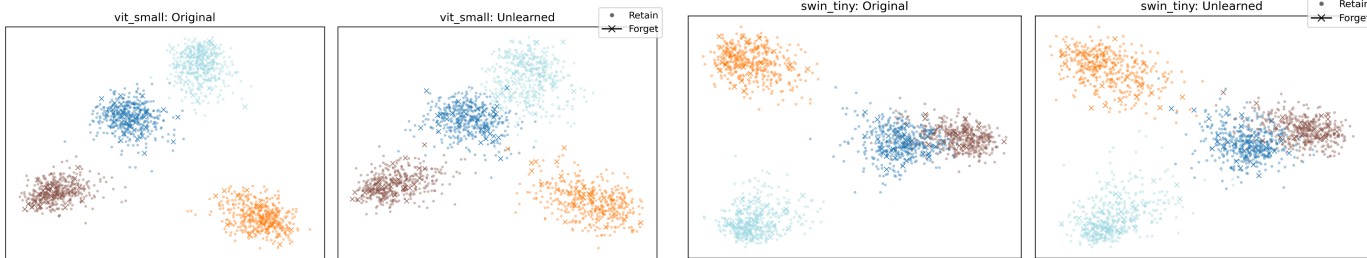

Figure 1: Representation visualization before vs after unlearning for ViT-Small and Swin-Tiny.

We can see that, before unlearning, clusters are more compact, with forget examples sitting within the clusters. After unlearning, we see clearly that: (i) forget examples are "being pushed out" of the core cluster; this is a strong indication that confidence of prediction is weakened for forget examples, which aligns with our quantitative Forget Accuracy and MIA results. (ii) remain examples stay largely within their core clusters, which aligns with our quantitative Retain Accuracy results.

## Continual Unlearning Performance

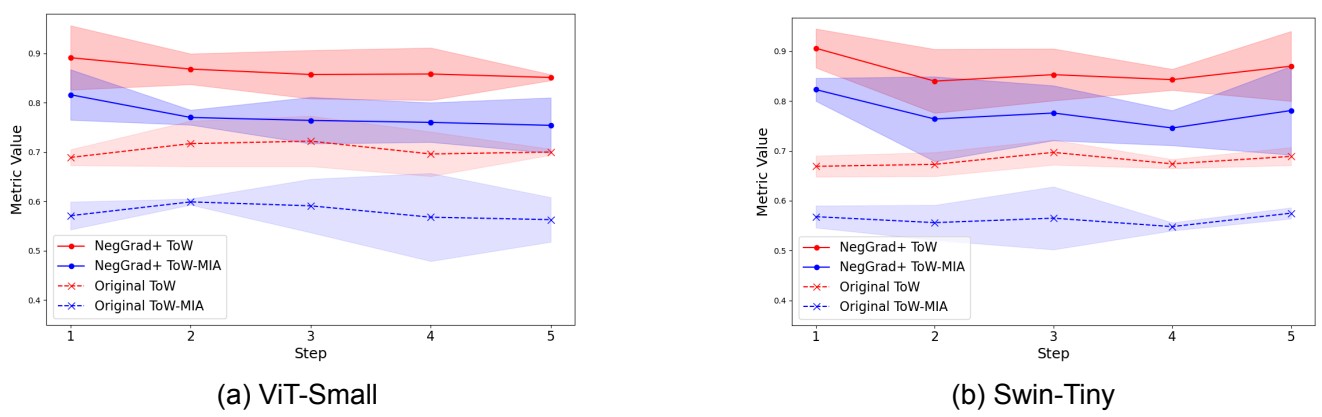

(a) ViT-Small

(b) Swin-Tiny

Figure 2: Continual Unlearning Performance Across 5 Steps on CIFAR-100 (updated from the paper's Fig. 2)

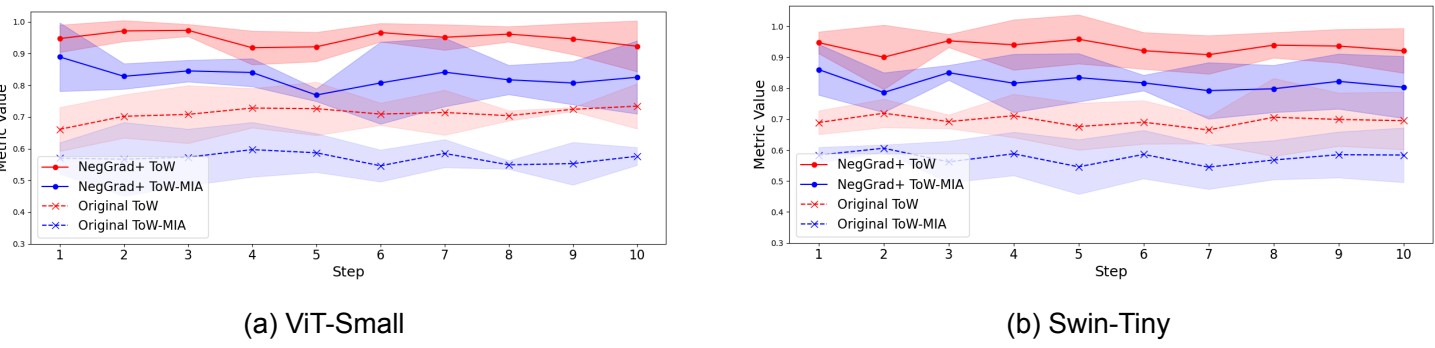

(a) ViT-Small

(b) Swin-Tiny

Figure 3: Continual Unlearning Performance Across 10 Steps on CIFAR-100