# OpenReview forum: "Benchmarking Machine Unlearning for Vision Transformers"
_ICLR.cc/2026/Conference — Submitted to ICLR 2026_

### Official Review · Reviewer_FuaG · 2025-10-18

**Soundness:** 3
**Presentation:** 4
**Contribution:** 3
**Rating:** 4
**Confidence:** 4

**Summary:**

This paper introduces the first systematic benchmark for machine unlearning (MU) in Vision Transformers (VTs), addressing a gap in the literature where prior work focused primarily on CNNs, LLMs, and diffusion models. The authors evaluate a set of MU algorithms (Fine-tune, NegGrad+, and SalUn with RUM meta-framework) across multiple VT architectures (ViT, Swin-T), varying model capacities, and datasets (CIFAR-10/100, SVHN, and ImageNet). The study also explores memorization dynamics, proxy validity, and continual unlearning performance, proposing a unified evaluation framework using ToW and ToW-MIA metrics. The benchmark reveals several insights: VTs exhibit memorization patterns similar to CNNs; CNN-derived memorization proxies (notably Confidence and Holdout Retraining) remain effective for VTs; NegGrad+ generalizes best across settings, and pretraining enhnces unlearning on simpler tasks.

**Strengths:**

- **Novelty and importance:** The work tackles a clear and timely gap in machine unlearning research as ViTs are increasingly central to modern vision systems. The inclusion of multiple VT families (ViT, Swin-T) and capacity scales demonstrates thoughtful coverage of architectural diversity.
- **Comprehensive experimental design:** Evaluates across four datasets of varying complexity and multiple model capacities, providing a nuanced understanding of how data scale and architecture interact with unlearning. Inclusion of both single-shot and continual unlearning is commendable, as continual settings reflect realistic deployment conditions.
- **Methodology:** The ToW and ToW-MIA metrics offer a unified and interpretable framework balancing forget quality and performance retention, building on and extending recent unlearning evaluation practices. Detailed hyperparameter configurations, reproducibility links, and confidence intervals enhance the transparency of the paper.
- **Insightful findings:** The paper provides actionable empirical insights: (a) pretraining mitigates over-memorization (b) Swin-T is more amenable to gradient-based unlearning and (c) proxies like Holdout Retraining scale well computationally. It also identifies non-trivial architecture–algorithm compatibilities, such as fine-tuning working better for ViT and NegGrad+ excelling for Swin-T.
- **Benchmarking Contribution:** Offers a publicly available benchmark and codebase that can serve as a foundation for future research, similar to TOFU or MU-Bench in other modalities.

**Weaknesses:**

- **Limited algorithmic diversity:** While the selected algorithms are representative, the benchmark excludes newer VT-specific unlearning methods (e.g., NOVO [1]), even if contemporaneous. Including at least a comparative baseline or partial replication would strengthen relevance.
- **Missed baselines from VT-specific continual learning (CL).** The study benchmarks MU algorithms but does not adapt or compare against strong CL methods purpose-built for Vision Transformers, such as lifelong/continual ViTs and exemplar-free approaches that rely on attention-, functional-, or weight-regularization [2,3]. Many of these techniques are replay-free (matching the data-deletion spirit) and could be repurposed as unlearning operators (e.g., using regularizers to suppress influence of $D_f$ while retaining $D_r$ ). Without these baselines, it’s hard to tell whether MU-specific methods truly outperform CL-style regularizers in VT settings. This gap is especially relevant given the paper’s emphasis on continual unlearning scenarios.
- **Focus on memorization-based methods** Along a similar note, the study’s restriction to memorization-leveraging algorithms provides depth but limits generalizability. It would be infurmative to see how non-memorization-based or regularization-based MU methods perform in VTs.
- **Metric Interpretability and Sensitivity:** ToW and ToW-MIA, though unified, may be opaque to practitioners: the product-form metric can conflate tradeoffs between retention and forgetting. Ablating or weighting terms could clarify sensitivity.
- **Limited Theoretical Framing:** While I understand that the paper is primarily meant to be empirical, a theoretical discussion of why transformer attention patterns might affect unlearning differently (e.g., entanglement across heads/layers) would be extremely beneficial to readers.
- **Scalability:** Although the study includes ImageNet-scale data, it remains confined to relatively small VT variants (≤50M parameters). Real-world ViT-L/16 or Swin-B models might exhibit different behaviors, particularly for continual unlearning. Can the authors comment on this?


**References**

[1] Roy, Soumya et al. “NOVO: Unlearning-Compliant Vision Transformers.” (2025)

[2] Pelosin, Francesco et al. “Towards Exemplar-Free Continual Learning in Vision Transformers: an Account of Attention, Functional and Weight Regularization.” CVPRW (2022).

[3] Wang, Zhen et al. “Continual Learning with Lifelong Vision Transformer.” CVPR (2022).

**Questions:**

- **Proxy Generality:** Have you tested whether confidence or holdout retraining proxies remain stable across fine-tuning stages or different pretraining corpora (e.g., ImageNet-21K vs. ImageNet-1K)?
- **Algorithm Interactions:** How does RUM partitioning (low/medium/high memorization) affect convergence stability for transformers with high inter-layer attention mixing? Does partition ordering matter?
- **Continual Unlearning Robustness:** In continual unlearning, would retraining intermediate models periodically (vs. sequential finetuning) improve long-term stability, or is catastrophic forgetting already minimal?

---

> ### Author Response · Authors · 2025-11-21
>
> We are very happy to read reviewers’ FuaG praise regarding the novelty and importance of our contributions, the comprehensive experimental design, the methodology, the paper’s insightful findings, and the benchmarking contribution. Reviewer FuaG raises a few very interesting issues, which we are happy to address below.
>
> **Weaknesses**:
>
> **W1. On Limited algorithmic diversity**: “While the selected algorithms are representative, the benchmark excludes newer VT-specific unlearning methods (e.g., NOVO [1]), even if contemporaneous. Including at least a comparative baseline or partial replication would strengthen relevance.”
>
> **Response**:
> Thank you for raising this issue: As FuaG and our paper mentioned, *NOVO is contemporaneous work*. We agree that we should position against it explicitly and more prominently in the paper itself. However, NOVO addresses a fundamentally different unlearning problem than the one studied in our benchmark:
>
> - First, **NOVO operates on a different problem setting, which differs from the classical unlearning problem** addressed with our paper. Specifically, NOVO is an *unlearning-compliant architectural solution trained from scratch, simulating unlearning operations during training*. The **classical unlearning problem** for image classification is formulated as a **post-hoc intervention on pre-trained models**. Here, we are benchmarking CNN-based unlearning methods, which are for post-hoc unlearning applied to an already-trained model.
>
> - Second, **NOVO is designed for class/sub-class unlearning** and does not study **instance-based** unlearning (which is the more challenging and most popular formulation of the problem studied in the MU image-classification literature). We benchmark MU methods for instance-based unlearning, which dominate the SoTA in the published MU literature.
>
> - Putting these together, NOVO cannot deal with instance-level unlearning as their unlearning simulations during training cannot possibly account for all possible combinations of sets of instances that an unlearning operation may specify after training - this is more (only) tractable for unlearning (sub)classes.
>
> For these reasons, we felt that including methods like NOVO as baselines would add extra dimensions to the problem setting, making results more brittle and less interpretable.
> Nonetheless, to address FuaG’s valid point, **we will more prominently discuss NOVO and related works, raising the important differences (brought up by this comment)** in the problem formulations and explaining our benchmarking setup in more detail.
>
> **W2.  Missed baselines from VT-specific continual learning (CL)**. “… Without these baselines, it’s hard to tell whether MU-specific methods truly outperform CL-style regularizers in VT settings. This gap is especially relevant given the paper’s emphasis on continual unlearning scenarios…”
>
> **Response**:
>
> The reviewer’s point is insightful: It may very well be possible to ‘reduce’ the problem of CL in VTs to solve MU in VTs, as hinted by FuaG. However, CL is **a different field of study**. There exists already ongoing work that attempts to blend these into a single solution for continual learning and unlearning. **Our paper is about benchmarking CNN-based MU solutions for VTs** under a unified protocol and metrics (and deriving relevant insights and new knowledge), **rather than studying "whether MU-specific methods truly outperform CL-style regularizers in VT settings and/or how to derive the latter"**. That is a different (albeit relevant) problem.
>
> In more detail: There exist fundamental arguments that point to a possible “impedance mismatch” when aiming to adopt CL solutions for MU. Particularly with respect to: (i) **objective and guarantees** (i.e., stability–plasticity vs unlearning/forgetting measured against retrain-from-scratch and with additional privacy MIA constraints); and  (ii) **granularity** (CL focuses on task/partition-level importance, whereas MU in instance-level unlearning). These raise difficult challenges when adopting/adapting or developing and evaluating CL solutions for MU.
>
> NB: Without a doubt, this issue is important. And it is also a current blind spot in the SOTA knowledge in the general area of MU (for VTs and beyond). Albeit largely orthogonal to our efforts/aims, **we will address this concern** in the final version, explicitly referring to this discussion, citing related works from CL, their possible adoption/adaptation for MU, and the relevant issues from above, as FuaG suggests.

---

> ### Author Response · Authors · 2025-11-21
>
> **W3. Focus on memorization-based methods**. “…Along a similar note, the study’s restriction to memorization-leveraging algorithms provides depth but limits generalizability.”
>
> **Response**:
>
> FuaG is correct: We had to tame expectations of a deep-dive analysis against generalization. We feel that a benchmarking paper (like ours) should strive primarily for deep and specific analyses, and our focus on memorization-leveraging approaches reflects/achieves this.
>
> Having said that, our aim was to give CNN-based MU algorithms the best possible chance of succeeding in VT environments. This hinges on employing their strongest possible variants. Prior work has shown that algorithms instantiated within the RUM framework can achieve this goal: Specifically, RUM  provides a principled way to boost a diverse set of MU approaches (e.g., gradient-based (NegGrad+), finetuning-based (FT), saliency-based (SalUn), and sparse-parameter-based (L1-Sparse)), by using a memorization-guided partitioning [Zhao et al., 2024]. Incorporating these methods within RUM has consistently been shown to improve their performance. Because this strategy relies on leveraging memorization behavior, we also examined how memorization patterns in VTs compare to those in CNNs and whether common proxies remain reliable in VT architectures (see Section 4.1). Our findings confirm that these proxies apply well to VTs, providing further justification for employing memorization-leveraging MU methods in this benchmark.
>
> Furthermore, to concretely and quantitatively address this comment, we **ran additional experiments, pitting the RUM-NG+, RUM-FT, and RUM-SalUn against the (vanilla) NG+, FT, and SalUn algorithms** (i.e., outside RUM). The experiments were performed on the CIFAR-100 dataset for both ViT-small (Table 1) and Swin-Tiny (Table 2). The results show that, across both architectures, the RUM-enhanced versions consistently outperform the vanilla ones, even in VT settings. This substantiates our above ‘qualitative’ responses. The results are presented in the tables below.
>
> Table 1: Performance with and without RUM on CIFAR-100 using ViT-Small.
> | Method     | **Methods within RUM** (results from Table 11.c) |          | **Methods outside RUM (new results)** |          |
> |------------|-------------------------|----------|-------------------------------|----------|
> |            | ToW (↑)      | ToW-MIA (↑) | ToW (↑)                      | ToW-MIA (↑) |
> | Original   | 0.619          | 0.538    | Same as left columns  |
> | Fine-tune  | 0.855          | 0.889    | 0.807             | 0.844    |
> | NegGrad+   | 0.931           | 0.838    | 0.888            | 0.845    |
> | SalUn      | 0.903        | 0.709    | 0.884           | 0.686    |
>
> Table 2: Performance with and without RUM on CIFAR-100 using Swin-Tiny
> | Method     | **Methods within RUM** (results from Table 11.d)  |            | **Methods outside RUM (new results)** |            |
> |------------|-------------------------|------------|-------------------------------|------------|
> |            | ToW (↑)                | ToW-MIA (↑) | ToW (↑)                      | ToW-MIA (↑) |
> | Original   | 0.670                  | 0.586      | Same as left columns|
> | Fine-tune  | 0.822                  | 0.839      | 0.774                        | 0.781      |
> | NegGrad+   | 0.975                  | 0.902      | 0.851                        | 0.831      |
> | SalUn      | 0.870                  | 0.751      | 0.833                        | 0.640      |
>
> [1] Zhao, Kairan, et al. "What makes unlearning hard and what to do about it." Advances in Neural Information Processing Systems 37 (2024): 12293-12333.
>
> **W4. Metric Interpretability and Sensitivity** : “ToW and ToW-MIA, though unified … can conflate tradeoffs between retention and forgetting. Ablating or weighting terms could clarify sensitivity”.
>
> **Response**:
>
> We completely agree. We have performed such analyses, and the results are available in **Appendix (A.4)**, where we present tables with the **per-term metrics** (retain accuracy, forget accuracy, test accuracy) and MIA for different methods/architectures. Thus, **our evaluation goes beyond showing the average (product) scores** and allow readers to inspect each component directly. For example, the CIFAR-100 ViT-Small results (Table 13.(c)) lists **Retain Acc / Forget Acc / Test Acc / MIA** for Retrain, Fine-tune, NegGrad+, and SalUn, under both proxies.
> We do understand that reviewers should not be expected to read Appendices. We would be happy to transfer any of these results to the main paper. We would sincerely appreciate FuaG’s suggestions/guidance for this, if they think it necessary.

---

> > ### Comment · Reviewer_FuaG · 2025-11-24
> > **results with and without RUM**
> >
> > Thank you for the additional experiments and results, given that RUM adaptively applies unlearning methods to forget set partitions arranged by memorization scores of instances, is it a surprise then that RUM-enhanced methods perform better here? Also, I do agree with the authors' take on algorithmic diversity and the continual learning link.

---

> > > ### Author Response · Authors · 2025-11-24
> > >
> > > Dear reviewer FuaG,
> > >
> > > Thank you for your continued engagement, for appreciating our new results, and for agreeing with our arguments on algorithmic diversity and continual learning, (which we derived in response to your valuable comments -- thank you).
> > >
> > > Your question in your comment above, "is it a surprise then that RUM-enhanced methods perform better here?" iles at the crux of what we are doing with this paper: We are answering exactly whether such memorization-based partitioning of the forget set, that is largely beneficial for CNNs, continues to be beneficial for VTs as well.
> > >
> > > Given that VTs have different architectures than CNNs (due to different attention mechanisms, etc.) the answer to the question is:
> > > * Not immediately clear and deserves a comprehensive study on its own (which we perform here). We call it a "pleasant surprise".
> > > * Relevant, as it would be great to show that CNN-based algorithms can be deployed within VTs, yielding very strong performance.
> > > * Significant, as we now have a very strong set of baselines:
> > >      * for future research to compare against.
> > >      * And, for practitioners who now have a great starting point: They need not wait for the development of VT-specific MU algorithms to be developed, which are very much lacking at this time, and for the field to mature, etc.: They can take ready, off-the-shelf, MU methods and start with them!
> > >
> > > We remain at your disposal with any other concerns/reservations you may have, that woulcd convince you to raise your evaluation score.

---

> ### Author Response · Authors · 2025-11-21
>
> **W5.  Limited Theoretical Framing**: “While I understand that the paper is primarily meant to be empirical, a theoretical discussion … would be extremely beneficial ..”.
>
> **Response**:
>
> We agree. However, we respectfully wish to point out that this is not just an ‘empirical research paper’ – it is a “benchmarking” paper. To our knowledge, benchmarking papers are only rarely, if ever, accompanied by theoretical analyses/framing.
>
> **W6. Scalability beyond ≤50M params and longer continual MU**.
>
> **Response**:
>
> FuaG is correct. Please note that we have already examined the effect of scaling of models to MU method performance, including various-sized models for both Swin and ViT (ranging from ViT-Tiny with  ~4M to Swin-Small with ~50M parameters). Alas, practical reasons forced us not to extend the study to even bigger scales as FuaG points out.
>
> Nonetheless, to respond to this concern, **we ran additional experiments with  Swin-B (~88M parameters)** on our bigger ImageNet-1k dataset (50K validation split, described in Sec. 4.4). This allows a direct comparison with Swin-Small (reported in Table 4 in our paper). Results for the Swin family are shown in the table below.
>
> Table 3: IN-1k data, on Swin-Base using HR proxy.
> | Method     | ToW for Swin-Base (88M) (new results) | ToW for Swin-Small (50M) (Table 4)|
> |------------|--------------------------|---------------------------|
> | Fine-tune  | 0.808 ± 0.007            | 0.780 ± 0.023             |
> | NegGrad+   | 0.837 ± 0.004            | 0.819 ± 0.018             |
> | SalUn      | 0.746 ± 0.013            | 0.743 ± 0.027             |
>
> We **also ran experiments for a bigger model from the ViT family** (ViT-B/16). The following table shows results for ViT-B/16 (with ca. **85M parameters** and patch size 16).
>
> Table 4: IN-1k data, on ViT-B/16 using HR proxy.
> | Method     | ToW for ViT-Base |
> |------------|------------------|
> | Fine-tune  | 0.843 ± 0.016    |
> | NegGrad+   | 0.842 ± 0.005    |
> | SalUn      | 0.773 ± 0.027    |
>
> We observe that performance appears to be slightly improved on the bigger Swin model (as would be expected given larger capacity). And the trends we report persist: FT and NG+ remain strong performers (with SalUn still comparatively weaker) for ViT-B as well. Overall, our conclusions and trends drawn from smaller models and datasets continue to hold for these bigger models of both families.
> In addition, we **extended the horizon of our continual MU study** to 10 steps, as FuaG suggests. The results are given below in Table 5.
>
> Table 5: Continual MU for 10 steps on CIFAR-100 using HR proxy.
>
> | Step    | **ViT-Small** ToW (↑)       | **ViT-Small** ToW-MIA (↑)   | **Swin-Tiny** ToW (↑)       | **Swin-Tiny** ToW-MIA (↑)   |
> |---------|------------------------------|------------------------------|------------------------------|------------------------------|
> | Step 1  | 0.947 ± 0.043                | 0.889 ± 0.108                | 0.947 ± 0.035                | 0.860 ± 0.082                |
> | Step 2  | 0.971 ± 0.033                | 0.828 ± 0.040                | 0.900 ± 0.104                | 0.786 ± 0.064                |
> | Step 3  | 0.973 ± 0.019                | 0.845 ± 0.034                | 0.953 ± 0.021                | 0.850 ± 0.024                |
> | Step 4  | 0.918 ± 0.053                | 0.840 ± 0.244                | 0.940 ± 0.081                | 0.816 ± 0.094                |
> | Step 5  | 0.921 ± 0.046                | 0.769 ± 0.020                | 0.958 ± 0.079                | 0.834 ± 0.138                |
> | Step 6  | 0.966 ± 0.029                | 0.807 ± 0.129                | 0.921 ± 0.059                | 0.817 ± 0.025                |
> | Step 7  | 0.951 ± 0.040                | 0.841 ± 0.108                | 0.908 ± 0.062                | 0.792 ± 0.111                |
> | Step 8  | 0.961 ± 0.024                | 0.817 ± 0.046                | 0.939 ± 0.041                | 0.798 ± 0.126                |
> | Step 9  | 0.946 ± 0.049                | 0.807 ± 0.068                | 0.936 ± 0.054                | 0.822 ± 0.089                |
> | Step 10 | 0.923 ± 0.080                | 0.825 ± 0.115                | 0.921 ± 0.072                | 0.803 ± 0.100                |
>
> The results confirm our earlier finding: there continues to exist at least one method-proxy pair (NG+ with HR) for which continual MU shows no substantial accumulated degradation in performance.

---

> ### Author Response · Authors · 2025-11-21
>
> **Questions:**
>
> **Q1: On proxy stability**: *Have you tested whether confidence or holdout retraining proxies remain stable across fine-tuning stages*
>
> In our benchmark, models are first fully trained (fine-tuned end-to-end) on the full training set before any MU is applied. We computed proxy values and their correlations on this final, pre-unlearning model. In the continual MU setting, we recompute proxies after each unlearning step so partitioning always reflects the *current* model state, avoiding dependence on earlier fine-tuning stages. We did not specifically benchmark the stability of proxy ranks across fine-tuning epochs. This is an interesting direction, albeit orthogonal to our aim of a post-hoc MU benchmark on fully fine-tuned VTs.
>
> **Q2: Algorithm Interactions**: *How does RUM partitioning (low/medium/high memorization) affect convergence stability for transformers with high inter-layer attention mixing? Does partition ordering matter?*
>
> RUM partitioning order does matter, as demonstrated in prior work [Zhao et al., 2024]. We followed those findings and use the recommended low->medium->high memorization ordering. Because our results show that memorization patterns in VTs are very similar to those in CNNs, we expect this ordering to remain effective, and our experiments confirm strong performance under this choice. Whether different tweaks (such as partition ordering, algorithmic blending etc.) can yield even better performance is an open question. Again, We emphasize that our benchmark  does not aim to find/derive the best MU algorithm for VTs, but instead to benchmark, analyze, and derive insights as to whether and when CNN-based SoTA and baselines can be employed directly for VTs, and if their performance is as good as that in CNNs.
>
> [1] Zhao, Kairan, et al. "What makes unlearning hard and what to do about it." Advances in Neural Information Processing Systems 37 (2024): 12293-12333.
>
>
> **Q3: Continual Unlearning Robustness**: *In continual unlearning, would retraining intermediate models periodically (vs. sequential finetuning) improve long-term stability, or is catastrophic forgetting already minimal?*
>
> We assume that by  ‘catastrophic forgetting’, FuaG refers to lowering the accuracy for remaining (not unlearned) instances. Our continual unlearning results show that this does not occur under the settings studied.
>
> That said, in principle, if one continuously applies MU operations, eventually the support (decision boundaries) learned for instances and classes by the model may be eroded, especially if large portions of the data are removed or if the removed instances are critical to class support. So both remain-accuracy and test-accuracy will be negatively affected. These are open questions for the MU literature as a whole, and we do not have a definitive answer for them.
>
> **Summary:**
> Again, we sincerely thank FuaG for their thoughtful comments. We believe we have addressed all concerns with qualitative arguments and with quantitative arguments, running additional experiments for the most critical issues raised. Please let us know of any outstanding issues.

---

### Official Review · Reviewer_ZpV7 · 2025-10-21

**Soundness:** 3
**Presentation:** 4
**Contribution:** 2
**Rating:** 4
**Confidence:** 4

**Summary:**

This paper proposes the first comprehensive benchmark of vision transformers (VT) in machine unlearning, which is commonly dominated by CNNs and LLMs. The work evaluates two vision transformer architectures (ViT and Swint-T) with two parameter sizes on established unlearning benchmarks (i.e., CIFAR-10/100, SVHN, and an ImageNet subset). Additionally, this work compares CNN and VT memorization capabilities, the influence of pretraining in model unlearning, and the performance stabilities under continual unlearning. Results highlight fundamental similarities between CNNs and VTs' unlearning behaviors and memorization patterns, while showing discrepancies for some unlearning methods.

**Strengths:**

1. The paper is well written and easy to understand.
2. The idea is interesting, and I believe the research question is well-justified.
3. The paper also explores memorization and continual unlearning, which are important aspects for the narrative.
4. I appreciated that the source code was attached to the submission.

**Weaknesses:**

**Major weaknesses**
1. **Limited findings.** While the idea of benchmarking vision transformers in machine unlearning is an interesting research direction, I believe the major finding of this paper is that VTs closely resemble CNN when unlearning, with some exceptions that I believe end up in the background. For instance, the usefulness of pretraining for VTs unlearning on small datasets is of low interest, considering that retraining is feasible for such small datasets. Therefore, I am unsure whether this paper's findings are sufficiently interesting for the ICLR community, considering that ViTs were already employed by a few unlearning papers [1, 2, 3, 4].
2. **Odd unlearning scheme.** This paper follows the unlearning scheme of Zhao et al. (2024). In a nutshell, it uses proxy measures to estimate sample memorization. Then, it exploits the estimated memorization to partition forget set samples into three subsets, i.e., low-, medium-, and highly-memorized samples. Unlearning is performed by sequentially unlearning these three subsets from low- to highly-memorized. Although this is a valid unlearning procedure, it is not widely adopted in the literature, to the best of my knowledge. Furthermore, it conditions unlearning on an arbitrary proxy measure for estimating memorization, partially hiding the real unlearning contribution. Therefore, I believe unlearning in a more "standardized" way would have at least strengthened this paper's claims, which appear limited to this specific scenario in my opinion.
3. **Limited number of models, model sizes, and datasets.** As this paper proposes to benchmark vision transformers in machine unlearning, I expect more architectures (e.g., segmentation models, object detection models, self-supervised models, or just more classification models), bigger model sizes (i.e., not limited to 50M parameters), and larger datasets. With such small models and datasets, this paper's findings have a limited impact, as it is unclear whether they can also be observed across multiple models, model sizes, and large datasets.

**Minor weaknesses**
1. It is unclear what $A$ represents in Eq. (3).
2. The "Memorization Proxy Metric" paragraph requires more context. For instance, it is unclear how ranking is performed, and the meaning of proxy could be better explained.
3. Pretraining advantage of VTs is not investigated on CNNs; therefore, the overall finding is inconclusive (i.e., this could also be true for CNNs).
4. It is unclear from Figure 2 whether the unlearning performance is good or bad without showing the original model ToW. Also, in this case, it would be interesting to compare with CNNs; otherwise findings are inconclusive.
5. I suggest adding TOW, TOW-MIA, and the original model to Table 13.
6. The paper references multiple times results that are in the appendix. This makes everything hard to follow as it requires the reader to constantly jump from the main paper to the appendix. If a figure is necessary for the main paper's narrative, I suggest including it in the main paper.

[1] Foster, Jack, Stefan Schoepf, and Alexandra Brintrup. "Fast machine unlearning without retraining through selective synaptic dampening." AAAI, 2024.\
[2] Chundawat, Vikram S., et al. "Can bad teaching induce forgetting? unlearning in deep networks using an incompetent teacher." AAAI, 2023.\
[3] De Min, Thomas, et al. "Unlearning personal data from a single image." TMLR, 2025.\
[4] He, Zhengbao, et al. "Towards natural machine unlearning." TPAMI, 2025.

**Questions:**

1. Are the results presented in Figures 3 and 4 averaged over multiple datasets? If not, then the claim in L.257-261 is not supported.
2. What is the rationale behind using the RUM unlearning scheme?
3. Could not the L.451-452 key takeaway be imputed to the retaining steps in NegGrad+?
4. Table 8 experiments show that for low and medium-memorized samples, the authors used 5 unlearning epochs, while for highly-memorized ones, they used 10. Does an epoch imply an entire iteration of the subset of the forget set and the entire retain set? If so, then the unlearning cost is just half of the original training cost (see Table 7).

**Motivations for my score**\
Despite the limited breadth of the benchmark, I believe the main findings are not surprising, nor do I think they are sufficiently interesting to be published on ICLR. Nonetheless, I am willing to reconsider my score based on the opinions of the other reviewers and the outcome of the rebuttal.

---

> ### Author Response · Authors · 2025-11-21
>
> We thank reviewer ZpV7 for their thoughtful review and constructive criticism. We are happy to read that they appreciated the paper’s presentation quality, the interesting problem studied, our benchmarking effort, and the additional insights regarding memorization issues in VTs and for continual MU in VTs. Below, we respond to the criticisms raised by ZpV7 in detail using qualitative arguments and quantitative arguments based on results from new experiments we performed.
>
> **Weaknesses:**
>
> **W1. “Limited findings / not sufficiently interesting”**:  “VTs closely resemble CNN when unlearning,…”
>
> **Response**.
> Our results go way beyond the conclusion that “VTs closely resemble CNNs when unlearning.” Specifically:
>
> - **On memorization and proxies in VTs**. We **measure (rather than assume)**, and show that CNN-based memorization proxies transfer to VTs. Both Confidence and Holdout Retraining proxies show strong, significant correlations with Feldman memorization on both ViT and Swin (Tables 1, 10). This empirically validates that memorization-aware MU, which is previously developed for CNNs, can be meaningfully extended to VTs. This is a new, interesting finding which is of interest in its own right.
>
> - **Method-architecture pairings and performance differences**. Fine-tune is shown to be consistently strong on ViT, while NegGrad+ is stronger on Swin-T. SalUn attains high ToW but is poor w.r.t. ToW-MIA for VTs. These effects are validated as stable across datasets, model architectures, and model capacities (Tables 2-4, 11; Figs. 1/5). This shows a nuanced picture that should be of interest to MU practitioners when deciding on algorithms for specific architectures, etc. And, notably, this goes against ‘standard’ results in the general MU literature where new algorithms are presented as uniformly superior across architectures, etc., without such nuanced differences.
> We show that simple(r) MU algorithms (like FT and NG+) work excellently in VTs, and FT works better/worse than NG+ under different architectures (ViT vs Swin, likely due to their different attention mechanisms). This finding reveals an interesting, nuanced big picture, which is new and significant for real-world MU adaptation.
>
> - **Pretraining and scale**. We reveal for the first time the “pretraining helps VTs” story. And we show that it holds on simpler data (e.g., CIFAR-10), but this effect weakens at higher complexity (e.g., CIFAR-100/ImageNet). We provide detailed ToW and ToW-MIA performance information to support this. This adds another piece of useful and actionable VT-specific novel and nuanced knowledge.
>
> - **Continual MU stability on VTs**. We analyze **continual MU** (several unlearning steps) under ToW/ToW-MIA for ViT and Swin, and provide per-step results (Table 14), in addition to the aggregate curves (Figure 2,6). Again, this yields practical insights for real-world settings where unlearning is performed periodically.
>
> - Finally, in addition to the above, we respectfully believe that ZpV7 is **underestimating the overall major finding**. It is not that “VTs closely resemble CNNs when unlearning”, but rather that **we can achieve strong performance for MU on VTs by using CNN-based MU algorithms**. This is new, actionable knowledge and is of real-world importance for MU deployment.

---

> ### Author Response · Authors · 2025-11-21
>
> **W2. “Odd unlearning scheme; proxy-conditioned; not standard”.**
>
> **Response**.
>
> Respectfully, this is not an “odd unlearning scheme”. Memorization-leveraging has been
> shown to improve MU performance for many SOTA MU algorithms [Zhao et al., 2024].
>
> - Our aim was to give CNN-based MU algorithms the best possible chance of succeeding in VT environments. This hinges on employing their strongest possible variants. Prior work has shown that algorithms instantiated within the RUM framework can achieve this goal: Specifically, RUM  provides a principled way to boost a diverse set of MU approaches (e.g., gradient-based (NegGrad+), finetuning-based (FT), and saliency-based (SalUn)), by using a memorization-guided partitioning [Zhao et al., 2024]. Incorporating these methods within RUM has consistently been shown to improve their performance. Because this strategy relies on leveraging memorization behavior, we also examined how memorization patterns in VTs compare to those in CNNs and whether common proxies remain reliable in VT architectures (see Section 4.1). Our findings confirm that these proxies apply well to VTs, providing further justification for employing memorization-leveraging MU methods in this benchmark. And, we note that this analysis of VT memorization and proxy behavior is itself a useful contribution, as reviewer ZpV7 agreed.
>
> - Additionally, the paper associated with RUM is not an “odd” outlier. It is a fairly new paper (late 2024), which may explain why it has not been so widely adopted as of yet. Our choice of using RUM is based on its demonstrated empirical benefits across diverse MU methods.
>
> - To further address the reviewer’s concern, we ran **additional experiments**, pitting the RUM-NG+, RUM-FT, and RUM-SalUn against the vanilla NG+, FT, and SalUn algorithms (i.e., without using RUM). The experiments were performed on CIFAR-100 on ViT-small and Swin-Tiny. These provide a reference point vs the RUM versions of each of these algorithms. The results show that across all cases, incorporating the algorithms within RUM yields better performance. These results complement the published results for CNN models, and exemplify and validate our choice for using the RUM memorization-leveraging framework. The results are presented in Tables 1 and 2 below (the best performing cases are marked in bold):
>
> Table 1: Performance with and without RUM on CIFAR-100 using ViT-Small.
> | Method     | **Methods within RUM** (results from Table 11.c) |          | **Methods outside RUM (new results)** |          |
> |------------|-------------------------|----------|-------------------------------|----------|
> |            | ToW (↑)      | ToW-MIA (↑) | ToW (↑)                      | ToW-MIA (↑) |
> | Original   | 0.619          | 0.538    | Same as left columns  |
> | Fine-tune  | **0.855**          | **0.889**    | 0.807             | 0.844    |
> | NegGrad+   | **0.931**           | 0.838    | 0.888            | **0.845**    |
> | SalUn      | **0.903**        | **0.709**    | 0.884           | 0.686    |
>
> Table 2: Performance with and without RUM on CIFAR-100 using Swin-Tiny
> | Method     | **Methods within RUM** (results from Table 11.d)  |            | **Methods outside RUM (new results)** |            |
> |------------|-------------------------|------------|-------------------------------|------------|
> |            | ToW (↑)                | ToW-MIA (↑) | ToW (↑)                      | ToW-MIA (↑) |
> | Original   | 0.670                  | 0.586      | Same as left columns|
> | Fine-tune  | **0.822**                  | **0.839**      | 0.774                        | 0.781      |
> | NegGrad+   | **0.975**                  | **0.902**      | 0.851                        | 0.831      |
> | SalUn      | **0.870**                 | **0.751**      | 0.833                        | 0.640      |
>
> [1] Zhao, Kairan, et al. "What makes unlearning hard and what to do about it." Advances in Neural Information Processing Systems 37 (2024): 12293-12333.

---

> ### Author Response · Authors · 2025-11-21
>
> **W3. “Limited models/sizes/datasets”**
>
> **Response**:
>
> - Our benchmark already includes two VT families (ViT and Swin), multiple model sizes, and three datasets (CIFAR-10, CIFAR-100, SVHN), as well as a larger ImageNet-scale setup (50k split) with Swin-Small.
>
> - We wish to politely push back on the **criticism that the benchmark should go beyond classification tasks** (such as detection, segmentation, etc.). To our knowledge, **all related SoTA MU papers for image tasks (including the four papers cited by the reviewer) report results only on classification**. To our knowledge, there is no established MU formulation or benchmark for segmentation or detection, making direct comparison infeasible. Classification therefore remains the standard and appropriate setting for a VT MU benchmark.
>
> Nonetheless, the reviewer is well justified to ask for checking if results hold for even larger models. To address this, we conducted additional experiments with Swin-B (~88M parameters) on our larger ImageNet-1k setting (described in Sec. 4.4). This allows us to directly compare the new results against the old results (in Table 4 in our paper, showing results for Swin-Small). Results are shown in the table below.
>
> Table 3: IN-1k data, on Swin-Base using HR proxy.
> | Method     | ToW for Swin-Base (88M) (new results) | ToW for Swin-Small (50M) (Table 4)|
> |------------|--------------------------|---------------------------|
> | Fine-tune  | 0.808 ± 0.007            | 0.780 ± 0.023             |
> | NegGrad+   | 0.837 ± 0.004            | 0.819 ± 0.018             |
> | SalUn      | 0.746 ± 0.013            | 0.743 ± 0.027             |
>
> We also **ran additional experiments with a bigger model from the ViT family** (ViT-B/16 (with ca. **85M parameters** and patch size 16)).  The following table shows these results.
>
> Table 4: IN-1k data, on ViT-B/16 using HR proxy.
> | Method     | ToW for ViT-Base |
> |------------|------------------|
> | Fine-tune  | 0.843 ± 0.016    |
> | NegGrad+   | 0.842 ± 0.005    |
> | SalUn      | 0.773 ± 0.027    |
>
> We observe that performance appears to be slightly improved on the bigger Swin model (as would be expected given larger capacity). And the trends we report persist: FT and NG+ remain strong performers (with SalUn still comparatively weaker) for ViT-B as well. Overall, our conclusions and trends drawn from smaller models and datasets continue to hold for these bigger models of both families.
>
> **Minor weaknesses:**
>
> (1) *Symbol in Eq. (3) unclear.*
>
> Thank you for pointing this out. We will add a clear explanation of the symbol and update the notation in the revised version.
>
> (2) *“Memorization Proxy Metric” needs context (ranking).*
>
> We give the Spearman protocol and ranking in Eq. (6) and App. A.2.2. We will gladly add two clarifying sentences into §3.1 and briefly define “proxy” on first use.
>
> (3) *Pretraining advantage not investigated for CNNs.*
>
> This is an interesting issue. However, as our focus is on applying the CNN-based MU algorithms on VTs, and the latter typically follow a pretrain-then-finetune paradigm, our benchmark simply tried to isolate this impact with respect to VTs only. Studying pretraining benefits for MU in CNNs is beyond this paper’s scope (and this issue is currently unexplored within the CNN MU literature).
>
> (4) *Figure 2 lacks Original model ToW; CNN comparison requested”.*
>
> Re: Original model ToW: Thanks for the suggestion. We agree that this will provide a nice reference point. We will add the ToW numbers for the Original VT model (before unlearning, i.e., step 0) to Figure 2, as requested.
>
> Re: CNN comparison: Our work focuses specifically on understanding continual unlearning (CU) behavior in Vision Transformers, with the key question being whether CU degrades MU performance in VTs. Our novel findings answer this in the negative. Adding results on how CU affects performance in CNNs is outside the scope of our benchmark. We therefore wish to keep the scope and focus to VTs, if possible.
>
> (5) *Add ToW/ToW-MIA and Original to Table 13.*
>
> Table 13 currently provides the components (retain acc/forget acc/test acc/MIA), and Table 11 provides the ToW and ToW-MIA results, including those for the original model. We could add these to Table 13, but we are concerned that this may make Table 13 overly complex. We are open to the reviewer’s preference and will adjust accordingly.
>
> (6) *Too many appendix references.*
>
> We understand this concern and unfortunately, this is a typical issue faced by all of us, given the length limitations. We look to the reviewer for guidance as to which parts to promote to the main paper, and we will gladly comply.

---

> ### Author Response · Authors · 2025-11-21
>
> **Questions:**
>
> **Q1.** *Are Figs. 3–4 averaged over multiple datasets?*
>
> No. Fig. 3 corresponds to CIFAR-100; Fig. 4 corresponds to CIFAR-10. We will revise L.257–261 to state this explicitly and reference the corresponding dataset in each caption.
>
> **Q2.** *Rationale for using RUM?*
>
> Please see our detailed/comprehensive response to W2 above.
>
> **Q3.** *Could L.451–452 takeaway be due to NegGrad+ retaining steps?*
>
> This is very insightful.
> The knowledge we wanted to extract with this experiment is based on the following key question: **Is it possible that, if we use the most promising (MU method, proxy) pair, we can lay to rest any reasonable “fears” of accumulated degradation during continual unlearning**.  To demonstrate this, we chose NegGrad+ explicitly (as the reviewer’s conjecture suggests) because it employs a two-pronged approach (working on both retain and forget sets, explicitly trying to ensure forgetting while maintaining utility) and all our results point ot NegGrad+ being a high-performing MU method.
>
> In this sense, our result is “existential” in nature - not “universal”. But the finding is positive (and indirectly, we hope this will steer colleagues coming up with new MU methods to test their methods using the protocol we set up here).
>
> Taking this comment into account, **our key takeaway in this section should be rephrased**, from “Minimal (if any) degradation **occurs** under continual unlearning in VTs.” to
> Minimal (if any) degradation **can be achieved** under continual unlearning in VTs.
>
> **Q4.** *What does an “unlearning epoch” cover; is cost half of training?*
>
> Table 8 specifies the epochs for (low→med→high) partitions. One unlearning epoch iterates over $D_r$ plus the current $D_f$ partition. The unlearning epoch duration is approximately comparable to that of the training epoch. The key is that unlearning requires ~ one order of magnitude fewer epochs than full training. We will clarify this in the revision.
>
> **Summary:**
> We sincerely thank ZpV7 for their thoughtful comments. We believe we have addressed all concerns with qualitative arguments and with quantitative arguments, running additional experiments for the most critical issues raised. Please let us know of any outstanding issues.

---

> ### Comment · Reviewer_ZpV7 · 2025-11-24
>
> ## W1
> I thank the authors for the clarification. My criticism is not directed at the validity of this paper's findings; rather, I believe machine unlearning practitioners can get limited benefits from this paper's insights. Although I did not mention it in the original review, as a machine unlearning practitioner myself, I find that hyperparameter selection is the major issue in the current state of research. Showing, for instance, that some algorithms are more robust to the hyperparameter selection, compared to others, is to me of greater interest. Instead, showing that Gradient Ascent performs better on Swin can be somewhat interesting, but as it is an unbounded algorithm, I must carefully tune the learning rate, and in the real world, I cannot compute TOW and TOW-MIA. Furthermore, I wish approximate unlearning to be cheaper than full retraining. Thus, even if I could compute TOW for free, optimizing the learning rate to satisfy unlearning without destroying my model is more expensive than model retraining. Additionally, I agree that authors are the first to point out that pretraining helps VT unlearning. However, if its contribution is limited for big datasets, I do not see how this can help the current state of research, which assumes datasets and models to be too big for model retraining. I hope this clarifies my position.
>
> On a short note about the points raised by the authors: I like the narrative that "[...] against ‘standard’ results in the general MU literature where new algorithms are presented as uniformly superior across architectures, etc., without such nuanced differences." However, the authors should have included more published algorithms other than SalUN to support this claim.

---

> > ### Comment · Reviewer_ZpV7 · 2025-11-24
> >
> > ## W2
> > I thank the authors for providing results without the RUM framework. My original intention was not to undermine the claims made in the RUM paper. Instead, I am questioning the decision to use a relatively new framework that, to the best of my knowledge, has only been adopted in a few papers in the vision domain. At the same time, I still have to see an LLM-related unlearning paper using it (I did not check every work that cites RUM). RUM introduces an extra "hyperparameter", i.e., the memorization proxy metric, which I believe adds extra complexity without actually helping this paper's narrative. Therefore, it is not safe to assume that practitioners from now on will use the RUM framework. For this reason, I would have given priority to the classical (no memorization-based distinction) instance-wise unlearning, and instead provided additional results in RUM.

---

> ### Comment · Reviewer_ZpV7 · 2025-11-24
>
> ## W3
> I thank the authors for the additional results. I agree that detection and segmentation are not properly investigated in the literature and that classification is the main targeted problem. My criticism is more on the breadth of search, which I find limited in every direction (model architectures and sizes, algorithms, and data). I understand that machine unlearning is expensive due to the hyperparameter search (see above), but further exploring at least one of the proposed axes would have increased this paper's impact as a benchmark. Following research trends in LLM, exploring bigger models that go beyond the standard classification model sizes would have been more realistic. Alternatively, one could have also increased the pool of algorithms. Furthermore, except for ImageNet-1k, all used datasets are "toy-datasets" and two of them are very similar (CIFAR-10/100).
>
> About minor weaknesses:
> - 3-4. I agree that this paper is about VTs and not CNNs. Yet, without testing it on CNNs as well, I do not know whether this is a particular feature of VTs or whether it is independent of the architecture. This is not a major issue, but instead a suggestion on how to improve the presentation.
> - 6. Honestly, I believe it is very personal, and how to deal with it depends on the author's taste. I find the paper a bit worthy, and shortening some sentences may help. Also, section 3.1 is relatively useless, considering that almost all researchers in this field know about the investigated datasets, and section 3.2 is way too long, considering the contained information. Conclusions are also very long compared to the average paper. Another example is Figure 1. By reducing the bars' width, sharing the y-axis, and using the entire paper's width, it could probably fit in one row. Yet, these are suggestions based on my taste.

---

> > ### Comment · Reviewer_ZpV7 · 2025-11-24
> >
> > ## Q
> > **Q1.** Sorry, what I meant is whether they were averaged over multiple **seeds**.
> >
> > **Q4.** If an unlearning epoch takes about the same time as a training epoch, and considering that fine-tuning takes 50 epochs while unlearning takes 20 epochs (5+5+10), how come it is an order of magnitude lower?

---

> ### Author Response · Authors · 2025-11-27
>
> ## **Comment W1:**
> “…I believe machine unlearning practitioners can get limited benefits from this paper's insights… Hyperparameter selection is the major issue… Gradient Ascent is unbounded and needs careful LR tuning; in the real world I cannot compute TOW/TOW-MIA… tuning could be more expensive than retraining… pretraining helps VT unlearning but its contribution is limited for big datasets… including more published algorithms than SalUn would help.”
>
> **Response:**
>
> Thank you for clarifying your practitioner viewpoint. We agree that operational aspects (e.g., hyperparameter selection, tunability, and cost relative to retraining) are important. However, our benchmark was designed with a different primary goal: to help systematically uncover **what works on ViTs, when, and why**, across architectures/datasets, rather than to fully optimize the deployment costs on MU algorithms. Nonetheless, we believe the paper’s usefulness for practitioners can be further strengthened rather straightforwardly:
>
> **1. On hyperparameter robustness & defaults.**
>
> We agree that hyperparameter selection is a major open challenge in machine unlearning. Our work does not aim to solve hyperparameter optimization itself.Think of it as providing a necessary step towards it. We will make more prominent  **which algorithm-architecture combinations are promising**.This explicitly informs practitioners. To make this more explicit/actionable, we will add a short **“Practitioner’s Corner”** table, summarizing the key configurations/settings  (optimizer, LR, weight decay, epochs, early-stopping criteria, etc.) that worked reliably in our experiments.
>
> To address the *“unbounded GA”* risk comment, we will be explicit about the “bounded” variant we used in practice.
>
> Regarding the concern about **“computing ToW/ToW-MIA during deployment**”: we would like to clarify that these metrics are used **only for evaluation**, not as part of the unlearning procedure itself. This is fully aligned with standard practice across the MU literature
> (e.g., [4,5,6,8]), where methods are evaluated **post-hoc** against an **evaluation metric** so that the research community can benchmark algorithms on common ground. Any MU algorithm – once selected – still requires hyperparameter search for the target setting. Thus, the need for an evaluation metric does not disadvantage our study; it is simply the mechanism by which comparative performance is established in nearly all published MU work.
>
> **2.  On “Tuning is more expensive than retraining.”**
>
> We would like to clarify that tuning is a one-time cost, same as tuning the original training pipeline, and does not occur per unlearning request. Once an MU configuration is selected, our RUM-based variants require typically **20 epochs per request, compared to 50 epochs for a full retrain**. Thus, each subsequent unlearning request is substantially cheaper than retraining, especially at larger model or dataset scales. While we focus on studying unlearning behaviors rather than optimizing for efficiency, the per-request cost gap remains significant. For completeness, we conducted additional experiment to explicitly measure the wall-clock times of the MU algorithms (both the vanilla and their RUM versions) vs the time to retrain. The Table below provides these results. These show that even the RUM-based MU algorithms are faster than retraining by a factor greater than 3X.
>
> **Table 6**: Run Time Estimate for CIFAR-100 (seconds).
>
> | Method     | ViT-Small RUM | ViT-Small Vanilla | Swin-Tiny RUM | Swin-Tiny Vanilla |
> |------------|----------------|-------------------|----------------|-------------------|
> | **Retrain**   | 1300.996       | 1300.996          | 1288.794       | 1288.794          |
> | **Fine-tune** | 332.557        | 91.258            | 345.678        | 97.748            |
> | **NegGrad+**  | 395.764        | 152.093           | 397.407        | 141.845           |
> | **SalUn**     | 313.946        | 95.609            | 318.005        | 112.402           |

---

> ### Author Response · Authors · 2025-11-27
>
> **3. On “Pretraining benefit “small on big data”.**
>
> Our contribution here is revealing the benefits of pretraining **and** its possible limitations. We observe that pretraining’s gain diminishes as dataset complexity grows (and this nuance itself is an important and previously unreported insight.), but it still stabilizes unlearning (smaller utility loss, fewer spikes in RA/MIA). The takeaway for practitioners is therefore not that “pretraining solves everything,” but that *pretraining can help*. We have shown this to be the case for moderately complex data – a scenario present in many real applications. We hypothesize that the same conclusions will hold for larger scale/complexity datasets **when model capacity scales accordingly**, and our preliminary experiments below support this intuition.
>
> Specifically, our previous results (Table 2 in our paper) show that the pretraining’s benefits diminish when the dataset becomes more complex (CIFAR-10 → CIFAR-100). We therefore conducted additional experiments on CIFAR-100 using the larger ViT-B and Swin-B models, investigating whether it is the balance between dataset complexity and model capacity that determines whether pretraining is beneficial. As shown in Table 7 below, although all models are pretrained, the -Base models “recover” the pretraining benefits that were diminished in the smaller-scale (ViT-Small / Swin-Tiny), as evidenced by their higher $D_f$ accuracy of $\theta_r$. This supports our intuition, but we leave a systematic validation of this to future work.
>
> **Table 7:** $D_f$ accuracies (in %) of model $\theta_r$ across architectures on CIFAR-100
>
> | Architecture | Confidence          | Holdout Retraining      |
> |--------------|----------------------|--------------------------|
> | ViT-Small    | 69.322 ± 1.788       | 68.767 ± 3.828          |
> | Swin-Tiny    | 69.833 ± 1.242       | 70.267 ± 2.848          |
> | ViT-Base     | 78.099 ± 1.894      | 77.469 ± 1.952    |
> | Swin-Base    | 78.600 ± 0.790    | 79.678 ± 2.380          |
>
> At any rate, investigating and pointing out the impact of pretraining for MU in VTs is something that our work surfaces, making a mark for it so that all future research (and practitioners) should consider. We do agree this is a starting point, and investigating pretraining benefits at larger (data and model) scales is a valuable next step. We will explicitly clarify this positioning in the revision.
>
> **4. On “More published algorithms than SalUn.”**
>
> Our goal was not to create a leaderboard for MU in VTs. Instead, it was to study fundamentally different MU **families of algorithms** and how they interact with VT architectures. To that end, we carefully selected three representative MU approaches which offer fundamentally different approaches to MU and whose key characteristics can be found in the majority of published MU methods, albeit implemented differently, entailing the key ingredients used across many papers. Specifically:
>
> - **FT** [1]: an MU approach using only retain data.
>
> - **NegGrad+** [2]: a *“push–pull”* gradient method (gradient updates on both retain (pull) and forget (push) data. The same control pattern underlies Bad Teacher [3]), SCRUB [2], LetheViT [4], Natural MU [5]. These mainly differ in how they implement the “push” term.
>
> - **SalUn** [6]: an *”identify–then–dampen”* paradigm where, first, important parameters are located and then perturbed/dampened. Different implementations of this idea appear in SSD [7], L1-sparse [8], Random Label [9], etc.
>
> We show that our results/trends hold for their RUM versions (and for their vanilla version, outside of RUM), on VTs. Studying these three families lets us ask which mechanism pairs best with which VT architecture, rather than simply aim for a leaderboard. This focus on fundamental underpinning mechanisms provides insights that holds for the three important baselines we used here, but further hopefully they generalize across many published algorithms. We believe this perspective is valuable, as it helps identify structural reasons behind algorithm-architecture interactions without getting lost in method-specific hyperparameters or implementation details (which particularly agrees with this reviewer’s perspective, based on their comments on sensitivity and complexity of MU algorithms on hyperparameters and implementation details).
>
> Finally, and more importantly: while it is always beneficial to study as many algorithms as possible, **this does not prevent this study from surfacing the strong, novel and significant takeaway message** (especially for practitioners of MU in VTs): namely, **there exist algorithm-architecture combinations** that can achieve high performance on VTs (equal to or better than the corresponding performance on CNNs for which said algorithms were derived).
>
> We sincerely appreciate the reviewer’s push toward practitioner-focused needs. We hope the clarifications and suggested edits above make the paper more useful to practitioners.

---

> ### Author Response · Authors · 2025-11-27
>
> ## **Comment W2:**
> “I thank the authors for providing results ... My original intention was not to undermine the claims made in the RUM paper… I am **questioning the decision to use a relatively new framework** … has only been adopted in a few papers in the vision … still have to see an **LLM-related unlearning paper** using it. **”RUM introduces an extra "hyperparameter"**, i.e., the memorization proxy metric, which I believe adds extra complexity without actually helping this paper's narrative. Therefore, **it is not safe to assume that practitioners from now on will use the RUM** framework. For this reason, I would have given priority to the classical (no memorization-based distinction) instance-wise unlearning, and instead provided additional results in RUM.”
>
> **Response:**
>
> **On RUM being a relatively new framework**. Thank you for raising this point. RUM is indeed a recent framework, and many MU papers appearing in 2025 may simply not have been aware of it during their design phase. We included RUM because prior work has shown consistent and substantial improvements across a broad range of MU algorithms. We do not wish or aim to be apologists for RUM itself, but it provides the best framework for our purposes. Since our goal is to study how CNN-based MU mechanisms transfer to ViTs, **it would be inappropriate to employ methods that systematically have been found to underperform**. Hence, our selection of the RUM framework.
>
> We agree with the reviewer that the original results raised questions w.r.t. whether the “classical” MU approaches would make a difference. This is why, in addition to the RUM-enhanced variants reported in the paper, prompted by this reviewer,  we included results for the classical (non-memorization-based) versions in our rebuttal. Our result show that the RUM versions continues to yield consistent performance improvements even when transferred to VTs.
>
> **Regarding RUM’s additional “hyperparameter”**: this corresponds to selecting a memorization proxy, and our experiments show that several proxies work reliably. More importantly, this does not affect the conclusions of our study: our aim is to analyze how different MU mechanisms transfer to ViTs, not to optimize for hyperparameter tuning, and these mechanism-level trends remain the same regardless of whether RUM is used. Nonetheless, we do provide to practitioners concrete suggestions showing proxies that reliably offer strong performance.
>
> Regarding the comment on **RUM for LLM MU**: the absence of RUM in LLM unlearning papers is expected and **does not reflect its unsuitability for vision MU**. LLM MU operates under a fundamentally different problem formulation (autoregressive objectives, sequence-level memorization, and distinct unlearning definitions and different memorization definition), and therefore uses different mechanisms and metrics than MU for vision classifiers. Thus, the fact that RUM has not appeared in LLM MU is not surprising, given the fundamental differences between the two settings. So, this does not by any means diminish RUM’s relevance, applicability, or significance in the vision-classification setting.

---

> ### Author Response · Authors · 2025-11-27
>
> ## **Comment W3: Limited models/sizes/datasets**
> “I thank the authors for the additional results. I agree that detection and segmentation are not properly investigated … and that classification is the main targeted problem. My criticism is more on the breadth of search, which I find limited  … Following research trends in LLM, exploring bigger models that go beyond the standard classification model sizes would have been more realistic. Alternatively, one could have also increased the pool of algorithms. Furthermore, except for ImageNet-1k, all used datasets are "toy-datasets" and two of them are very similar (CIFAR-10/100).”
>
> **Response:**
>
> We appreciate the reviewer’s suggestion to broaden the experimental design. However, we would like to note that our study already incorporates significant breadth, which we respectfully feel is  not fully reflected in the reviewer’s comments.
>
> **On model sizes:** Our study already spans a wide and meaningful range of VT capacities, from a few million parameters (ViT-Tiny) to 20+ million (ViT-Small, Swin-Tiny), to 40+million (Swin-Small) and up to 80+million parameters (ViT-B and Swin-B). This covers both major VT families (ViT and Swin) across small, medium, and larger variants commonly used in vision classification research, and our conclusions hold consistently across these scales. Given finite compute resources, we believe this diversity offers significant insights (if not more) than devoting compute resources to simply testing extremely large models.
>
> **On dataset sizes:** Our dataset choices directly follow established practice in MU for image classification. Nearly all prior works in this domain evaluate on CIFAR-10/100, (maybe) SVHN, and some ImageNet variant/subset [1,2,3,4,5,6,7,8,9]. We therefore followed this practice here as well. This enables apples-to-apples comparisons and helps isolate and identify architecture-method effects.
>
> **On the request to follow LLM trends:** We would like to respectfully push back on the suggestion to “follow LLM trends” and use models beyond standard classification sizes: MU in LLM is a fundamentally different problem setting. The underlying architectures, training objectives, memorization definitions, and MU mechanisms & metrics differ substantially between LLMs and vision classifiers. Pushing to model sizes beyond those used for vision/classification tasks would (i) conflate settings and reduce comparability and explainability with existing MU-for-classification results, and (ii) fall outside the scope of our study, which is explicitly focused on MU for vision classifiers.
>
> **On “more algorithms”:** Please see our response to W2 above. In short, with our benchmark we chose to seek/emphasize depth and mechanism clarity in the standard classification MU setting, and surface actionable guidance, as explained in our previous response. We believe this focus best contributes to  the current SoTA understanding as it applies to VTs as opposed to studying segmentation/detection, multimodal/SSL setups, and very-large-scale models (which, albeit, clearly deserve future research focus).
>
> **On minor weaknesses.**
> We appreciate the suggestion to compare with CNNs directly and will note this in the limitations section. We also thank the reviewer for the presentation/style suggestions and will shorten Sections 3.1 and 3.2, condense the conclusion, and consider tightening Figure 1 in the revised version to add space for reporting the new results and advice to practitioners.

---

> > ### Author Response · Authors · 2025-11-27
> >
> > ## **Questions:**
> > Q1. *Sorry, what I meant is whether they were averaged over multiple seeds.*
> >
> > Yes, they were averaged over multiple seeds. Following the procedure of Feldman & Zhang [10], we compute memorization scores by **training hundreds of models, each initialized with a different random seed**. An example’s memorization is then estimated by comparing its probability when included in training vs. when excluded, aggregated across these models, so the final results are averaged over these hundreds of seeds. This is the established proper  methodology for studying memorization in vision models, and we adopt it directly.
> >
> > Q4. *If an unlearning epoch takes about the same time as a training epoch, and considering that fine-tuning takes 50 epochs while unlearning takes 20 epochs (5+5+10), how come it is an order of magnitude lower?*
> >
> > Thank you for raising this. The question here concerns efficiency, and we clarify as follows:
> >
> > 1. RUM is chosen for its strongest ToW-based performance, not efficiency per se.
> > RUM typically uses more epochs than vanilla MU variants, and this is an expected trade-off . Our goal here is to evaluate the *high-performing* MU variants on VTs, not to directly optimize for minimal compute overheads.
> >
> > 2. We also include vanilla FT / NG+ / SalUn in our rebuttal results, all of which use only 5 epochs. These are cheaper than RUM, but are consistently weaker in performance. Including both versions makes the performance-efficiency trade-off more explicit.
> >
> > 3. Efficiency also depends on dataset scale. On modest datasets like CIFAR, even retraining from scratch is inexpensive, so efficiency differences are less pronounced. At larger scales, e.g., ImageNet-level datasets or larger VT models, we expect the difference between 5-20 unlearning epochs (either vanilla or RUM) and full retraining becomes much more substantial. However, we note again that **efficiency is not the primary focus of this work**. Our aim is instead to compare unlearning behavior across architectures and mechanism families. For completeness, we provide below wall-clock times, to illustrate the relative costs. we intentionally do not focus on large-scale efficiency benchmarking, as that would detract from the main scientific questions we wish to address.
> >
> > Table 6 below shows the actual wall-clock times on CIFAR-100 for both full retraining and our unlearning runs. The results show that even on this modest dataset, unlearning runs (FT, NegGrad+, SalUn) are noticeably cheaper than full retraining; and RUM, while more expensive than its vanilla counterparts, still remains significantly cheaper per-request than a full retrain, by a factor of >3X.
> >
> > **Table 6: Run Time Estimate for CIFAR-100 (seconds)**
> >
> > | Method     | ViT-Small RUM | ViT-Small Vanilla | Swin-Tiny RUM | Swin-Tiny Vanilla |
> > |------------|----------------|-------------------|----------------|-------------------|
> > | **Retrain**   | 1300.996       | 1300.996          | 1288.794       | 1288.794          |
> > | **Fine-tune** | 332.557        | 91.258            | 345.678        | 97.748            |
> > | **NegGrad+**  | 395.764        | 152.093           | 397.407        | 141.845           |
> > | **SalUn**     | 313.946        | 95.609            | 318.005        | 112.402           |
> >
> > [1] Warnecke, Alexander, et al. "Machine unlearning of features and labels." NDSS 2023.
> >
> > [2] Kurmanji, Meghdad, et al. "Towards unbounded machine unlearning." NeurIPS 2023.
> >
> > [3] Chundawat, Vikram S., et al. "Can bad teaching induce forgetting? unlearning in deep networks using an incompetent teacher." AAAI 2023.
> >
> > [4] Tong, Yujia, et al. "LetheViT: Selective Machine Unlearning for Vision Transformers via Attention-Guided Contrastive Learning." arXiv 2025.
> >
> > [5] He, Zhengbao, et al. "Towards natural machine unlearning." IEEE TPAMI 2025.
> >
> > [6] Fan, Chongyu, et al. "Salun: Empowering machine unlearning via gradient-based weight saliency in both image classification and generation." ICLR 2024.
> >
> > [7] Foster, Jack, et al. "Fast machine unlearning without retraining through selective synaptic dampening." AAAI 2024.
> >
> > [8] Jia, Jinghan, et al. "Model sparsity can simplify machine unlearning." NeurIPS 2023.
> >
> > [9] Golatkar, Aditya, et al. "Eternal sunshine of the spotless net: Selective forgetting in deep networks." CVPR 2020.
> >
> > [10] Feldman, Vitaly, and Chiyuan Zhang. "What neural networks memorize and why: Discovering the long tail via influence estimation." NeurIPS 2020.

---

> > > ### Comment · Reviewer_ZpV7 · 2025-11-27
> > >
> > > ## About hyperparameter selection
> > >
> > > I agree with the authors. Their work evaluates "what works on ViTs, when, and why, across architectures/dataset." I brought out hyperparameter selection as an example of findings that, in my view, are more interesting for the machine unlearning community. To this extent, I am fully aware of how ToW is computed and that it is used for evaluation only. Again, it was only used as an example for the argument above.
> > >
> > > What do the authors mean by a "bounded" variant of GA? Are they using NPO[1] or SimNPO [2]?
> > >
> > > [1] Zhang, Ruiqi, et al. "Negative preference optimization: From catastrophic collapse to effective unlearning." arXiv, 2024.\
> > > [2] Fan, Chongyu, et al. "Simplicity prevails: Rethinking negative preference optimization for llm unlearning." In NeurIPS, 2025.
> > >
> > > ## About more published algorithms
> > >
> > > Also, here, I agree with the authors. I brought this up because the author's claim that "[...] against ‘standard’ results in the general MU literature where new algorithms are presented as uniformly superior across architectures, etc., without such nuanced differences" is not supported without additional algorithms. Nonetheless, I am not questioning the choice of algorithms; rather, showing multiple algorithms' performance would have helped this paper.
> > >
> > > ## About the breadth of search
> > >
> > > I see the authors' point of view; yet, I still feel that this benchmark does not provide full coverage of the topic. Overall, my suggestion is not to jointly expand all of the mentioned axes. Instead, I believe that exploring at least one of these a bit further would have brought up more interesting findings.
> > >
> > > ## About the cost of unlearning
> > >
> > > I agree that the objective here is not towards efficiency, still, it is a bit high (not a major issue in my view). Finally, related to the answer above, showing unlearning performance at different compute budgets could have also been an interesting axis.
> > >
> > > ## Concluding remarks
> > >
> > > I thank the authors for providing such detailed responses to my questions. I will ponder my final rating based on the rebuttal outcome.

---

> > > > ### Author Response · Authors · 2025-11-27
> > > >
> > > > Dear reviewer,
> > > >
> > > > We have sincerely enjoyed our discussions and we thank you for this and especially your continued engagement!
> > > >
> > > > One final point on your last question: "What do the authors mean by a "bounded" variant of GA?".
> > > > This was in response to your comment: "Gradient Ascent performs better on Swin, but as it is an unbounded algorithm..."
> > > >
> > > > Our response was meant as a (polite) attempt to point out a possible misunderstanding: We never claimed that "Gradient Ascent performs better on Swin". It is NegGrad+ that was shown to perform better on Swin - that was our claim.
> > > > *NegGrad+ is not GA*. It is a version that explicitly tackles the "unboundedness" of GA by incorporating a push/pull (ascend/descend on forget/retain data). This is an algorithm (from NeurIPS23) that is shown to be very strong in all of our papers/research (despite the fact that it is not used as widely as it deserves as a baseline in many papers, which instead, for some reason, use only GA or FT as baselines, but not NegGrad+, which prudently aims to combine both...).
> > > >
> > > > And to complete adding usefulness to practitioners, we said in that response that we would be sharing the relevant additional details (eg about learning rates, decays, etc) so that practitioners enjoy a good starting point.
> > > >
> > > > Again, thanks a lot for your comments and engagement. We have tried hard to answer all your concerns with explanations and a suite of additional experiments/results that further substantiated our claims. Please let us know if there are any other questions.

---

### Official Review · Reviewer_wENs · 2025-10-26

**Soundness:** 4
**Presentation:** 4
**Contribution:** 4
**Rating:** 8
**Confidence:** 4

**Summary:**

This paper presents the first comprehensive benchmark of machine unlearning (MU) methods applied to Vision Transformers (ViTs), focusing on how well existing CNN-based unlearning algorithms perform in this new setting. The authors evaluate several representative methods (Fine-tune, NegGrad+, SalUn) across different ViT architectures (ViT, Swin-T), model capacities, datasets, and both single-shot and continual unlearning protocols. They found that ViTs and CNNs exhibit similar memorization behaviors, and that simple measures such as Confidence and Holdout Retraining can effectively assess memorization strength to guide the unlearning process. Among the tested methods, NegGrad+ consistently delivers the most robust unlearning performance, while Swin-T generally outperforms ViT on more complex datasets. Overall, the study provides a reproducible framework for evaluating MU in ViTs and highlights key factors, such as architecture, capacity, and memorization, that shape unlearning effectiveness.

**Strengths:**

The paper has the following strengths:
- It is the first systematic benchmark of machine unlearning methods applied to ViT, addressing a clear research gap. Furthermore, it offers a reproducible framework for this purpose
- the study uses multiple architectures (ViT, Swin-T), datasets of varying complexity, and both single-shot and continual unlearning setups, ensuring robust and generalizable findings.
- it provides valuable insights into how memorization patterns, model capacity, and architectural design influence unlearning effectiveness

**Weaknesses:**

The paper has the following weaknesses:
- While datasets vary in complexity, they remain classification-focused. Extending to other vision tasks (e.g., segmentation, detection) could have broadened the paper's impact.
- The paper relies heavily on quantitative evaluation. However, more qualitative examples or visual explanations could have made the findings more intuitive.

**Questions:**

Besides the weaknesses mentioned above, here are my other concerns:
- Extend the continual unlearning experiments beyond five steps to better assess long-term stability and cumulative degradation.
- Discuss possible causes of degradation (e.g., parameter drift, overfitting) and how they might be mitigated.
- Incorporate qualitative examples or visualizations showing how unlearning affects image representations or attention maps in ViT/Swin-T
- In the 'Limitations' section, discuss potential directions for extending the benchmark, e.g unlearning in self-supervised or multi-modal transformer models.

---

> ### Author Response · Authors · 2025-11-21
>
> We thank reviewer wENs for the thoughtful assessment and for praising our benchmark’s novelty, breadth, and insights. We address the weaknesses and questions below.
>
> **W1: “Only classification datasets; segmentation/detection could broaden impact.”**
>
> **Response.** We agree and **will make our paper’s scope more explicit** in the revision. But, please note there are important reasons for “only classification”.
>
> **Our benchmark employs the (standard) prevailing MU-for-vision setup**: post-hoc unlearning on classifiers (MU SoTA methods do not evaluate on segmentation/detection tasks). This choice was made for two reasons: First, to enable apples-to-apples comparisons with the SoTA CNN MU literature. Second, other tasks (like segmentation) require different unlearning guarantees (e.g., region-level forgetting, i.e., forgetting *parts/regions of an image*) and different evaluation designs. Furthermore, they typically utilize different architectures where  U-Net units are added to the standard VT architecture.
>
> **W2: “Heavily quantitative; more qualitative examples would help.”**
>
> **Response.**  Agreed, this would indeed be very helpful. We provide such examples in the supplementary material: visualizations for classes (clusters) and their remain examples (dots) and forget examples (crosses), before and after unlearning, for both ViT-Small and Swin-Tiny using NegGrad+ with the HR proxy.
>
> We can see that, before unlearning, clusters are more compact, with forget examples sitting within the clusters. After unlearning, we see clearly that: (i) forget examples are “being pushed out” of the core cluster; this is a strong indication that confidence of prediction is weakened for forget examples, which aligns with our quantitative Forget Accuracy and MIA results. (ii) remain examples stay largely within their core clusters, which aligns with our quantitative Retain Accuracy results.
>
> **Questions:** “Besides the weaknesses mentioned above, here are my other concerns:”
>
> **Q1.** *“Extend the continual unlearning experiments beyond five steps”*
>
> Thank you for this suggestion. We have **extended our experiments to address this concern**, using the same setting as before (NegGrad+ and HR) and evaluating both ViT-Small and Swin-Tiny. The results are given below in Table 5, showing results for 10 unlearning steps.
>
> Table 5: Continual MU with NG+ for 10 steps on CIFAR-100 using HR proxy.
> | Step    | **ViT-Small** ToW (↑)       | **ViT-Small** ToW-MIA (↑)   | **Swin-Tiny** ToW (↑)       | **Swin-Tiny** ToW-MIA (↑)   |
> |---------|------------------------------|------------------------------|------------------------------|------------------------------|
> | Step 1  | 0.947 ± 0.043                | 0.889 ± 0.108                | 0.947 ± 0.035                | 0.860 ± 0.082                |
> | Step 2  | 0.971 ± 0.033                | 0.828 ± 0.040                | 0.900 ± 0.104                | 0.786 ± 0.064                |
> | Step 3  | 0.973 ± 0.019                | 0.845 ± 0.034                | 0.953 ± 0.021                | 0.850 ± 0.024                |
> | Step 4  | 0.918 ± 0.053                | 0.840 ± 0.244                | 0.940 ± 0.081                | 0.816 ± 0.094                |
> | Step 5  | 0.921 ± 0.046                | 0.769 ± 0.020                | 0.958 ± 0.079                | 0.834 ± 0.138                |
> | Step 6  | 0.966 ± 0.029                | 0.807 ± 0.129                | 0.921 ± 0.059                | 0.817 ± 0.025                |
> | Step 7  | 0.951 ± 0.040                | 0.841 ± 0.108                | 0.908 ± 0.062                | 0.792 ± 0.111                |
> | Step 8  | 0.961 ± 0.024                | 0.817 ± 0.046                | 0.939 ± 0.041                | 0.798 ± 0.126                |
> | Step 9  | 0.946 ± 0.049                | 0.807 ± 0.068                | 0.936 ± 0.054                | 0.822 ± 0.089                |
> | Step 10 | 0.923 ± 0.080                | 0.825 ± 0.115                | 0.921 ± 0.072                | 0.803 ± 0.100                |
>
> The results show that our conclusions for continual MU continue to hold. Specifically, there continues to exist a pair of MU (method, proxy) (e.g. NG+ with HR) where continual MU does not result in substantial accumulated degradation of performance.

---

> ### Author Response · Authors · 2025-11-21
>
> **Q2.**  *Discuss possible causes of degradation (e.g., parameter drift, overfitting) and how they might be mitigated*.
>
> Thank you for this thoughtful question. We assume reviewer wENs refers to possible degradation of utility and/or of forgetting quality during continual unlearning (CU).
>
> This is a very interesting comment touching upon many intertwined decisions: the MU algorithm (e.g. FT vs NG+), the VT architecture (e.g., ViT vs Swin family, which adopt different approaches to attention), the memorization proxies used, etc. And within each possible combination of the above, there are many variables at play: the learning rate, decay strategies, whether treating retain phases differently than forget phases, etc. In regards to steps for mitigation, the above outlines the key levers: pairings of MU method, proxy, and architectures, learning rates, interventions wr.t. Retain accuracy vis-a-vis forget accuracy, proxies etc.  For this reason, we chose what our results showed to be a promising combination (NG+ with HR). And, NG+ was chosen exactly because it inherently minimizes/balances such degradations, as it offers a two-pronged solution (applying gradient ascent on forget and descent on retain data). This balances out and corrects any possible over-aggressive actions towards retain utility vs forget quality and vice versa.
>
> Albeit interesting/important, this line of investigation lies outside the scope of a single paper and deserves attention in its own right. Our aim with this paper was to **show that there exists a solution (combination of MU algorithm and proxy) that can keep the accumulation of degradation at bay during CU on VTs** (both on ViT and Swin-T). Taking this comment into account, **our key takeaway in this section should be rephrased**, from *“Minimal (if any) degradation **occurs** under continual unlearning in VTs.”* to  *”Minimal (if any) degradation **can be achieved** under continual unlearning in VTs”*.
>
> Beyond this, we could offer intuitions as to what are more specific “possible causes of degradation” but we wished to refrain from doing so as this issue deserves a separate, proper scientific study on its own. We would be happy to surface this discussion in our limitations section.
>
> **Q3.**  *Incorporate qualitative examples or visualizations showing how unlearning affects image representations or attention maps in ViT/Swin-T.*
>
> Please see our response to W2 above.
>
> **Q4.** *In the 'Limitations' section, discuss potential directions for extending the benchmark, e.g unlearning in self-supervised or multi-modal transformer models.*
>
> We will gladly add this discussion.
>
> **Summary:**
> We sincerely thank reviewer wENs for their thoughtful comments. We believe we have addressed all weaknesses and questions with qualitative arguments (visualizations) and with quantitative arguments, running additional experiments. Please let us know of any outstanding issues.

---

> > ### Comment · Reviewer_wENs · 2025-11-27
> > **Official Comment by Reviewer wENs**
> >
> > I read authors' rebuttal and all my concerns have been addressed. Therefore, I maintain my initial rating which was 'Accept'.

---

### Official Review · Reviewer_X2HB · 2025-10-27

**Soundness:** 3
**Presentation:** 3
**Contribution:** 3
**Rating:** 6
**Confidence:** 4

**Summary:**

This paper presents the first systematic benchmark for machine unlearning in Vision Transformers, which extend previous researches in CNNs, LLMs, and diffusion models. The authors benchmark three representative MU algorithms — Fine-tune (FT), NegGrad+, and SalUn — all within the RUM meta-framework that leverages memorization-based partitioning of the forget set. They evaluate these methods across two VT families with different model capacities on multiple dataset. Its insights that CNN-derived algorithms remain effective for VTs and that memorization proxies are transferable is novel

**Strengths:**

1. Novel benchmark addressing an unstudied area on the machine unlearning for Vision Transformers.
2. Comprehensive experimental validation across architectures, datasets, algorithms, and proxy metrics.
3. Novel findings that  CNN-based MU methods can transfer effectively to VTs.
4. Novel findings that highlighting the distinct unlearning behaviors of ViT and Swin-T.

**Weaknesses:**

1. The study relies on CNN-derived algorithms, with limited exploration of VT-specific unlearning methods. This might weaken the findings that "CNN-based MU methods can transfer effectively to VTs".
2. The downstream tasks are simply classification task, cannot provide these claims also works on segmentation tasks or deepth estimation tasks.

**Questions:**

1. If downstream tasks become more complex like segmentation or deepth estimation, will the findings in paper still work?
2. Can the observed memorization patterns extend to self-supervised pretrained VTs?
3. How the self-attention framework in transformers influence the MU and data memorization?
4. what are the insights behind the findings that ViT achieves stronger results with fine-tuning–based methods while Swin-T performs better with gradient-based unlearning approaches like NegGrad+.

---

> ### Author Response · Authors · 2025-11-21
>
> We are very happy to read the reviewer's praise regarding the novelty of our benchmark, the comprehensive experimental validation, the novelty of our findings with respect to that CNN-based MU methods can transfer successfully to VTs and with respect to the distinct behaviours of ViT and Swin-T.
> Reviewer X2HB raises interesting issues, which we are happy to address below.
>
> **Weaknesses**:
>
> **W1. “...Limited exploration of VT-specific algorithms”.**
>
> **Response**:
>
> This is a correct observation. The key reason for this, however, is exactly because, **at the time of the writing of this paper, no VT-specific MU algorithms existed**! This highlights and emphasizes even more the value of our work here. In other words, given the absence of VT-specific algorithms, can we utilize CNN-based algorithms for the task? Which CNN-based algorithm? for which VT architectures? Can we design and use a fair comparison field/benchmark to help in developing better VT-specific algorithms? etc.
>
> To our knowledge, a **contemporaneous** paper appeared in arXiv (in August, shortly before our ICLR26 submission in September) proposing a new VT-specific MU algorithm [Tong et al, 2025]. Thus, it could not be incorporated experimentally here. Following the ICLR author instructions, we cited and discussed it briefly (2nd paragraph of Section 2.2).
>
> Regardless and importantly, although Tong et al. compare against CNN-based algorithms, **it does not include the strongest CNN-based algorithms**. The current SoTA in MU for vision/classification tasks is set by the RUM framework [Zhao et al, 2024], which shows that the previous SoTA MU algorithms, if incorporated within RUM, can achieve better performance. Tong et al. use these earlier methods only in their vanilla form (which Zhao et al. already show that RUM can significantly boost their performance), and they omit other important baselines such as NegGrad+, which we include and show to be highly competitive.
>
> As a side note, there is another recent work that appeared in July 2025 [Roy et al, 2025], for a technique called NOVO, which we also cited and discussed in section 2.2. However, NOVO solves a different unlearning problem: it focuses on **class/subclass unlearning** and operates in a **training-time** setting. In contrast, our benchmark (and the broader MU literature for vision classification) studies **post-hoc, instance-level unlearning**, where the goal is to remove the influence of specific training instances *after* the model has been trained. Because of this difference, NOVO is not directly comparable to the methods evaluated in our work.
>
> We will be very happy to extend our discussion surrounding (the lack of) VT-specific MU algorithms and the role our benchmark can play, positioning against [Tong et al, 2025] and [Roy et al, 2025].
>
> [1] Tong, Yujia, et al. "LetheViT: Selective Machine Unlearning for Vision Transformers via Attention-Guided Contrastive Learning." arXiv preprint arXiv:2508.01569 (2025).
>
> [2] Roy, Soumya, et al. "NOVO: Unlearning-Compliant Vision Transformers." arXiv preprint arXiv:2507.03281 (2025).
>
> [3] Zhao, Kairan, et al. "What makes unlearning hard and what to do about it." Advances in Neural Information Processing Systems 37 (2024): 12293-12333.
>
> **W2:  “The downstream tasks are simply classification task, cannot provide these claims also works on segmentation tasks or depth estimation tasks”**.
>
> **Response**:
>
> X2HB is correct. But please note this choice follows the standard practice in the MU literature: **existing SoTA MU algorithms are put forth and shown to work on vision/classification tasks**, including all prominent methods (like SCRUB, NegGrad+, L1-sparse, SalUn, etc). So this is not a peculiarity of our setup and benchmark. In fact, we are aware of no published research on MU for depth estimation or for segmentation.
>
> Interestingly, the MU problem on segmentation adds salient twists (e.g., remove *parts/segments of an image*, such as the parts carrying spurious/biased regions). Such characteristics make the problem itself and the evaluation of MU algorithms completely different, which is beyond the scope of this work.
>
> Additionally, popular architectures for segmentation (e.g., U-Net variants with transformer components) differ significantly from the “pure” VT architectures studied here, which makes direct transfer of our findings non-trivial.

---

> ### Author Response · Authors · 2025-11-21
>
> **Questions:**
>
> **Q1.** *If downstream tasks become more complex like segmentation or deepth estimation, will the findings in paper still work?*
>
> **Answer:** It is difficult to make direct claims, since, as argued above, the MU problem changes substantially for such tasks. We are aware of (unpublished) work that shows the MU problem for segmentation tasks (e.g., for bias/spurious region removal) may become easier or harder based on the richness/correctness of annotations of image segments, which surfaces unique-to-the application features and reveals interesting trade-offs therein. And as stated earlier, segmentation introduces a different unlearning objective (e.g., forgetting regions rather than entire instances), different evaluation protocols, and typically different architectures (e.g., U-Net–style hybrids rather than pure VTs). As such, extending our findings to these tasks would require new problem definitions and a dedicated study.
>
> **Q2.** *Can the observed memorization patterns extend to self-supervised pretrained VTs?*
>
> **Answer:** Our paper’s scope includes only the standard pretrain-the-finetune setting. We do not claim transferability of our results into self-supervised (SS) settings. We will gladly add this clarification to our paper, as X2HB suggests.
>
> Having said that, the pretrain-then-finetune setting is the dominant setting for MU research. For reference, even recent (contemporary) papers (e.g. [Tong et al., 2025]) contributing VT-specific MU algorithms operate in this setting. Finally, MU operating in SS settings with the accompanying lack of labels, presents challenges not addressed by the current MU SoTA methods.
>
> **Q3 and Q4.** *How the self-attention framework in transformers influence the MU and data memorization? … And what are the insights behind the findings that ViT achieves stronger results with fine-tuning–based methods while Swin-T performs better with gradient-based unlearning approaches like NegGrad+.*
>
> **Answer:** This is a very interesting question. We have found that it comes down to the global vs hierarchical/windowed attention in these models. Empirically, we see a consistent architecture-best-method pairing that maps to the models’ attention “topologies”.
> ViTs with their global attention tend to diffuse/share information across heads and layers. So, in this setting, features are typically already well-separated. And even though information is diffuse, after finetuning, much of the decision-making lies with the classifier head and last blocks. Hence, the cheapest place to change classifier predictions is the classifier head (and last blocks). Thus, “forgetting” a subset mostly means moving the decision boundary slightly, which can be done by operating on the classifier head without rewriting earlier layers.
>
> On the other hand, in Swin models, attention is localized (within windows). Its attention is then aggregated hierarchically, which in turn creates many distinct pathways. Hence, gradient-based MU (NG+) tends to localize gradient updates to these distinct pathways, which are separated from others. This leads to tolerating gradient updates better, without damaging utility.
>
> A full forensic analysis of the above question and findings would be a great step for future work, and we would gladly add this statement to our paper.
>
> **Summary:**
> We hope we have addressed all concerns raised by the reviewer X2HB. Please let us know if there are any other concerns.

---

> > ### Comment · Reviewer_X2HB · 2025-11-24
> >
> > Thank you for your response. All of my concerns have been addressed. I will maintain my evaluation as weak accept.

---

> > > ### Author Response · Authors · 2025-11-24
> > > **Thank you for your engament.**
> > >
> > > Dear reviewer X2HB.
> > >
> > > Thank you for your response and for acknowledging that our response addressed all your concerns.
> > >
> > > We remain at your disposal with any other concerns/reservations you may have, whose answers will help you improve your 'weak accept' evaluation.

---

### Comment · Area_Chair_bX6o · 2025-11-24
**Please engage into the discussion with authors and fellow reviewers**

Dear reviewers,
The authors have already provided their responses. Do they address your concerns?
Please engage into the discussion with authors and fellow reviewers.
Thanks!
Best,
AC

---

### Author Response · Authors · 2025-12-03
**Final Comment to AC and All Reviewers**

We thank all reviewers for their careful reading, constructive feedback, and for engaging deeply with our work, as well as the original and new ACs for their engagement and efforts.

To aid the new AC on their task, below we summarise the discussion, the **clarifications we provided**, and the **several additional experiments/analyses** we performed to address reviewers’ concerns. We then list the explicit revisions we will incorporate into the revised version of the paper.

**TL;DR**. Reviewers consistently highlighted the **novelty and importance** of our work, noting that this is the *first systematic benchmark* of MU methods on Vision Transformers, which address a clear, timely gap and offers a reproducible framework and public codebase (X2HB, wENs, FuaG, ZpV7).
They praised the **comprehensive experimental design** across multiple VT families, model capacities, diverse datasets, and both single-shot and continual MU, yielding robust and generalizable findings (X2HB, wENs, FuaG). Reviewers also appreciated our methodology and insights (X2HB, wENs, FuaG, ZpV7), and commended the clarity and reproducibility of the paper, remarking that it is well written, easy to follow, and accompanied by detailed configs and source code (ZpV7, FuaG).

The original ratings were (4, 4, 6, 8) with all confidences set to 4. After our responses, 2 reviewers (wENs, and X2HB) explicitly stated that “**all concerns have been addressed**”, one reviewer (FuaG) explicitly mentioned that they **agree with our responses** on their key concerns and provided no additional concerns, and one reviewer (ZpV7) thanked us for “**such detailed responses**” we provided and they added no additional concerns.


### **1. In more detail: Summary of clarifications and new results provided during rebuttal**

**Scope of algorithms, tasks, and experimental setup.**
We clarified why we use FineTune/NegGrad+/SalUn as representative MU families (X2HB W1, ZpV7 W1 follow-up, FuaG W1) and why we focus on the standard MU setting, i.e., post-hoc instance-level unlearning for image classification (ZpV7 W3, FuaG W1), and why we do not extend to segmentation/detection, depth estimation, or self-supervised settings (X2HB W2/Q1/Q2, wENs W1, ZpV7 W3).
We discussed our experimental scope in terms of architectures, model sizes, and datasets, and added larger-scale experiments (Table 3,4 in the rebuttal) to further expand the experimental space (ZpV7 W3, FuaG W6). We also clarified technical details (memorization computation, proxy stability, RUM partition order, etc.) (ZpV7 Q1, FuaG Q1/Q2). No reviewer raised additional concerns after these clarifications.

**Main findings and contributions.**
We reinforced and refined the core contributions:
- Reaffirm that memorization & proxies transfer to VTs [X2HB Q3, ZpV7 W1]
- Consolidate method-architecture pairings and their interpretations [X2HB Q3, ZpV7 W1, FuaG W2/W3]
- Refine our conclusions on pretraining, supported by new ViT-B/Swin-B results (Table 7 in the rebuttal). [ZpV7 W1/follow-up]
- Extend continual unlearning to 10 steps (Table 5; Figure 3 in the Supplementary Material) [wENs Q1/Q2, FuaG W6] , and clarify its conceptual distinciton from continual learning [FuaG W2].
These responses were explicitly acknowledged by FuaG and wENs, and ZpV7 did not raise further objections.

**RUM vs vanilla MU.**
To address concerns about using a relatively new framework “RUM”, we added full comparisons of vanilla FT/NG+/SalUn vs their RUM variants (Table 1,2 in the rebuttal) [ZpV7 W2/Q2/follow-up, FuaG W3]. Both reviewers accepted these additional results and raised no further objections.

**Qualitative examples.**
We added qualitative visualisations (Figure 1 in the Supplementary Material) to show representation changes for forget vs retain examples before and after unlearning [wENs W2/Q3].

**Efficiency concerns.**
While efficiency is not the primary focus of this benchmarking paper, we provided wall-clock runtime measurements (Table 6 in the rebuttal) to address this concern [ZpV7 Q4/follow-up].

### **2. Summary of reviewers’ post-discussion statements**

- **Reviewer X2HB:** “All their concerns have been addressed.”
- **Reviewer wENs:** “All their concerns have been addressed.”
- **Reviewer FuaG:** “… agree with the discussion on algorithmic diversity and continual learning …”. No pushback or further concerns were raised.
- **Reviewer ZpV7:** “… thanks for “such detailed answers”. No pushback or further concerns were raised.

(The above statements appear explicitly in the discussion thread.)

---

> ### Author Response · Authors · 2025-12-03
>
> ### **3. Changes we will include in the camera-ready based on revoewr feedback and additional experimental results.**
>
> Below we list exact modifications, grouped by where they will appear.
>
> **Main Paper**
>
> (a) Updated figures/tables
>
> - Update Fig.2 to add “Original” ToW/ToW-MIA curve to the CU plot (as shown in supplementary materials for the rebuttal).
> - Tighten Fig.1 to save space
>
> (b) Revised pretraining discussion
>
> Clarify that pretraining helps but with diminishing benefits as dataset complexity increases, unless model capacity scales; add supporting text summarising our new findings in ViT-B/Swin-B experiments during rebuttal.
>
> (c) Clarify diversity of selected MU algorithms
>
> Add explicit justification that FT, NG+, SalUn cover the three major MU mechanisms (retain-only; push-pull gradient; identify-then-dampen).
>
> (d) Add actionable guidance for practitioners
> A short subsection with:
> - recommended architecture-method pairs (e.g., ViT / FT, Swin / NG+)
> - reliable proxy choices (Confidence / HR)
> - stable training recipes (LR, scheduling, partition order)
> - which proxy to choose under which scenario (e.g., HR/NG+ for CU).
>
> (e) Minor corrections
> - Clarify notation in Eq. (3).
> - Shorten Sections 3.1-3.2 and condense the Conclusion (ZpV7).
> - Update limitations paragraph to include future directions: segmentation/detection, self-supervised VTs, multimodal transformers.
>
> **Appendix (new or expanded sections)**
>
> (f) New Appendix section: ViT-B & Swin-B results
> - Full tables for large-model experiments (IN-1k split).
> - Discussion on capacity vs dataset complexity.
>
> (g) New Appendix section: RUM vs vanilla MU
>
> - Provide comparison tables for vanilla FT/NegGrad+/SalUn vs their RUM variants, to confirm that RUM continually improve MU in VTs and our conclusions still hold. We will also briefly summarised it in the main paper.
>
> (h) Expanded Appendix sections
> - Provide a clearer description of memorization score computing for Figs. 3-4 in section A.2.1.
> - Add detail on proxy definitions and ranking procedure in section A.2.2.
> - Extend continual-MU figures/tables with the 10-step results in section A.6.
> - Add a short discussion on CL-based approaches and why they remain orthogonal in section A.6.
>
> We thank all reviewers again for the constructive dialogue. We believe the additional experiments, clarifications, and the associated revisions significantly strengthen the paper and make its contributions clearer and more actionable for both researchers and practitioners.
>
> If further adjustments would be helpful for the AC or reviewers, we will be happy to incorporate them.

---

### Meta-Review · Area_Chair_y17o · 2025-12-30

**Summary:**

The paper performs a benchmark of unlearning algorithms on vision transformers (VTs). One reoccurring concern was about scaling, which the authors addressed by adding experiments for ith Swin-B (~88M parameters). The reviewers’ main concern is about the fact that while focusing on VTs, the paper uses hardly any VT specific analysis, namely no analysis of self-supervised training, only CNN-based unlearning algorithms, no tasks beyond classification, such as segmentation or depth estimation, no study of the impact of the attention mechanisms that are special to VTs. Additionally, they have concerns about the memorization-based unlearning evaluation, which is not particularly common in most unlearning literature (maybe apart from Zhao et al, 2024, which the authors continuously cite for validating their approach). The third reviewer summarizes the AC’s impression after the rebuttal very well: „I believe machine unlearning practitioners can get limited benefits from this paper's insights“. Therefore, the AC recommends rejecting the paper.

**Reviewer Concerns:**

The reviewers main concerns are about the fact that the paper has very little VT specific setup, experiments, insights. This concerns:
1. Algorithms: Here, the authors mention that the only existing mechanism for VT unlearning came out short before ICLR. It will probably be beneficial to evaluate it for a resubmission.
2. Purely supervised learning focus: The authors clarify that they only focus on the pretrain-finetune setup and do not provide further insights on how things would look for self-supervised learning.
3. No VT-specific tasks: The paper focuses on classification only, no segmentation or depth estimation. While the authors correctly summarize that these setups need other unlearning objectives, it leaves the VT-specific insights limited.
4. No analysis of VT-specific attention: While the authors provide some observations in the rebuttal, they suggest that a comprehensive study would be material for future work.

One reviewer wanted to get visual insights, and the authors described some cluster behavior, namely  that clusters are more compact, however, searching for that results in the paper (ctrl+F) for „cluster“ did not yield any results, and the AC could not find that in the paper. Yet, the reviewer seemed satisfied and maintained their score.

Another concern that has only been addressed by pointing to Zhao et al, 2024, was the focus on unlearning evaluation under different degrees of memorization, which, also according to the AC’s experience with LLM, Diffusion, and CNN unlearning, seems a rather unusual lens and approach.

The final concern is about the pure empirical nature of the work, which might be too much to ask for a benchmark paper. Maybe re-submitting to a dedicated benchmark track might prevent the paper from facing similar feedback in the future.

**Reviewer Scores:**

Based on the concerns and the interactions during the rebuttal, I do not expect the reviewers would have raised their scores.

---

### Decision · Program_Chairs · 2026-01-26

Reject